# SOX11 promotes epithelial/mesenchymal hybrid state and alters tropism of invasive breast cancer cells

Erik Oliemuller[1], Richard Newman[1], Siu Man Tsang[1], Shane Foo[2], Gareth Muirhead[1], Farzana Noor[1], Syed Haider[1], Iskander Aurrekoetxea-Rodríguez[3], Maria dM Vivanco[3], Beatrice A Howard[1]*

[1]The Breast Cancer Now Toby Robins Research Centre, The Institute of Cancer Research, London, United Kingdom; [2]Translational Immunotherapy Team, Division of Radiotherapy and Imaging, The Institute of Cancer Research, London, United Kingdom; [3]CIC bioGUNE, Basque Research and Technology Alliance (BRTA), Derio, Spain

**Abstract** SOX11 is an embryonic mammary epithelial marker that is normally silenced prior to birth. High *SOX11* levels in breast tumours are significantly associated with distant metastasis and poor outcome in breast cancer patients. Here, we show that SOX11 confers distinct features to ER-negative DCIS.com breast cancer cells, leading to populations enriched with highly plastic hybrid epithelial/mesenchymal cells, which display invasive features and alterations in metastatic tropism when xenografted into mice. We found that SOX11+DCIS tumour cells metastasize to brain and bone at greater frequency and to lungs at lower frequency compared to cells with lower SOX11 levels. High levels of SOX11 leads to the expression of markers associated with mesenchymal state and embryonic cellular phenotypes. Our results suggest that SOX11 may be a potential biomarker for breast tumours with elevated risk of developing metastases and may require more aggressive therapies.

*For correspondence: beatrice.howard@icr.ac.uk

Competing interests: The authors declare that no competing interests exist.

## Introduction

SOX11 is an embryonic mammary factor that is not expressed in normal breast epithelial cells after birth (*Wansbury et al., 2011*). However, *SOX11* is expressed in many triple negative and HER2+ invasive breast cancers (*Wansbury et al., 2011*). *SOX11* expression in invasive breast cancer is associated with increased distant metastasis formation (*Oliemuller et al., 2017*). Inhibition of *SOX11* by siRNA suppressed growth and proliferation of ER- breast cancer cell lines, but had no significant effect on growth and proliferation of ER+ breast cancer cell lines (*Shepherd et al., 2016*). *SOX11* repression using siRNA reduced both cell migration and invasion in basal-like breast cancer (BLBC) cell lines, supporting a role for SOX11 in promoting breast cancer progression. In addition, *SOX11* inhibition in MDA-MB-468, a BLBC line, resulted in reduced expression of *FOXC1*, *CCNE1*, *KRT14*, *MIA* and *SFRP1*, all of which are PAM50 genes that are highly expressed in BLBC. This finding suggests SOX11 may modulate key features of basal-like cancer cells, including Keratin 14 expression, a marker expressed by basal cells and some luminal cells within the terminal ductal lobular unit, where many breast cancers arise (*Gusterson and Eaves, 2018*).

Our previous studies showed that when constitutively expressed by the human breast epithelial cell line MCF10A, SOX11 increased the number of basal/myoepithelial clones formed in standard colony-formation assays and led to increased mammosphere formation, suggesting that SOX11 can modulate mammary progenitor features and stem cell activity when expressed in mature postnatal mammary epithelial cells (*Oliemuller et al., 2017*). Furthermore, we also showed that SOX11

promotes invasive transition of DCIS.com cells that were injected into the mammary duct to mimic formation of DCIS-like lesions within the mammary duct prior to progression to invasive state and tumour formation (*Oliemuller et al., 2017*). Expression of SOX11 in preinvasive breast lesions and potential sites of microinvasion in samples from DCIS cases supports a role for SOX11 in promoting in situ to invasive breast carcinoma transition. These findings led us to hypothesise that reactivation of embryonic mammary developmental programmes mediated by SOX11 in postnatal breast epithelial cells would promote invasive progression of pre-invasive DCIS breast lesions and acquisition of features associated with poor patient outcome, including formation of distant metastasis.

Therefore, to further explore the role of SOX11 in regulating cancer stem cell (CSC) states and to determine whether SOX11 promotes metastatic dissemination of invasive breast cancer cells, we have created an inducible model to study the roles of SOX11 in progenitor/stem cell regulation and breast cancer progression in vitro and in vivo. In the model described here, SOX11 is expressed at higher levels in DCIS.com cells driven from the EF1A promoter after induction with doxycycline (DOX), when compared to another model we have previously used to study DCIS progression, in which low levels of SOX11 expression is constitutively driven by the CMV promoter (*Oliemuller et al., 2017*). In the inducible model presented here, SOX11 expression confers epithelial/mesenchymal hybrid state, features typical of embryonic mammary cell phenotypes, and influences organ-specific metastatic tropism to breast cancer cells. Finally, we also identify several novel SOX11 targets, many of which are highly correlated with *SOX11* expression in primary breast cancers and breast cancer metastases.

## Results

### Inducible expression of SOX11 leads to changes in stem cell profiles of DCIS.com cells

To investigate the role of SOX11 in breast cancer progression, we used the pINDUCER21 system to stably transduce DCIS.com cells, an invasive cell line from the MCF10A breast cancer progression series, so that SOX11 was expressed only when induced with Doxycycline (DOX) (referred to as iSOX11 cells) (*Figure 1A–B*). The results show a significantly higher, sustained expression of SOX11 levels compared with the previous constitutive model we have used to study DCIS progression which lost SOX11 expression over time (*Figure 1—figure supplement 1*; *Oliemuller et al., 2017*). As expected, SOX11 localised mostly to the nuclei in iSOX11 cells, similar to that observed in SOX11+ DCIS case samples (*Figure 1A–C* and *Figure 1—figure supplement 1*). SOX11 is also detected in the cytoplasm of iSOX11 cells using western blotting (*Figure 1A*), a location that was not observed in the DCIS-SOX11 cells (data not shown), showing that some differences exist when SOX11 is expressed at different levels in the two models.

To study the role of SOX11 in regulating stem cell state, we used the inducible iSOX11 model and assessed CSC profiles. CSCs are subpopulations of cancer cells sharing similar characteristics as normal stem or progenitor cells such as self-renewal ability and multi-lineage differentiation to drive tumour growth and heterogeneity. ALDH1 and CD24 are widely used CSC markers in breast cancer (*Liu et al., 2014*). A higher proportion of CD24+ and ALDH+ cells are detected in iSOX11 cells compared to control iEV cells (DCIS.com cells stably transduced with pINDUCER21 empty vector and induced with DOX) by flow cytometry (*Figure 1D–E* and *Figure 1—figure supplements 1–2*). Moreover, CD24 expression levels were significantly increased in a DOX dose-dependent manner in iSOX11 cells (*Figure 1—figure supplement 1*). CD24 is predominantly located in the membrane and CD24 was detected in the cytoplasmic fraction of both iEV and iSOX11 cells, and also detected in the nuclear fraction of iSOX11 cells, which has been previously reported in breast cancer (*Figure 1F*; *Duex et al., 2017*).

When cells were grown on collagen, we detected loss of membranous E-Cadherin localisation and acquisition of N-Cadherin expression in iSOX11 cells and formation of bud-like structures (*Figure 1G–H*). Nuclear expression of N-Cadherin has been reported previously, primarily in poorly differentiated breast tumours (*Rezaei et al., 2012*). We also detected a significant increase in iSOX11 cells expressing Vimentin by western blotting; increased Vimentin was detected in both the cytoplasmic and nuclear fraction. Vimentin was not observed in the nucleus of cells grown in the same conditions when stained by immunofluorescence, suggesting that the nuclear fraction contains

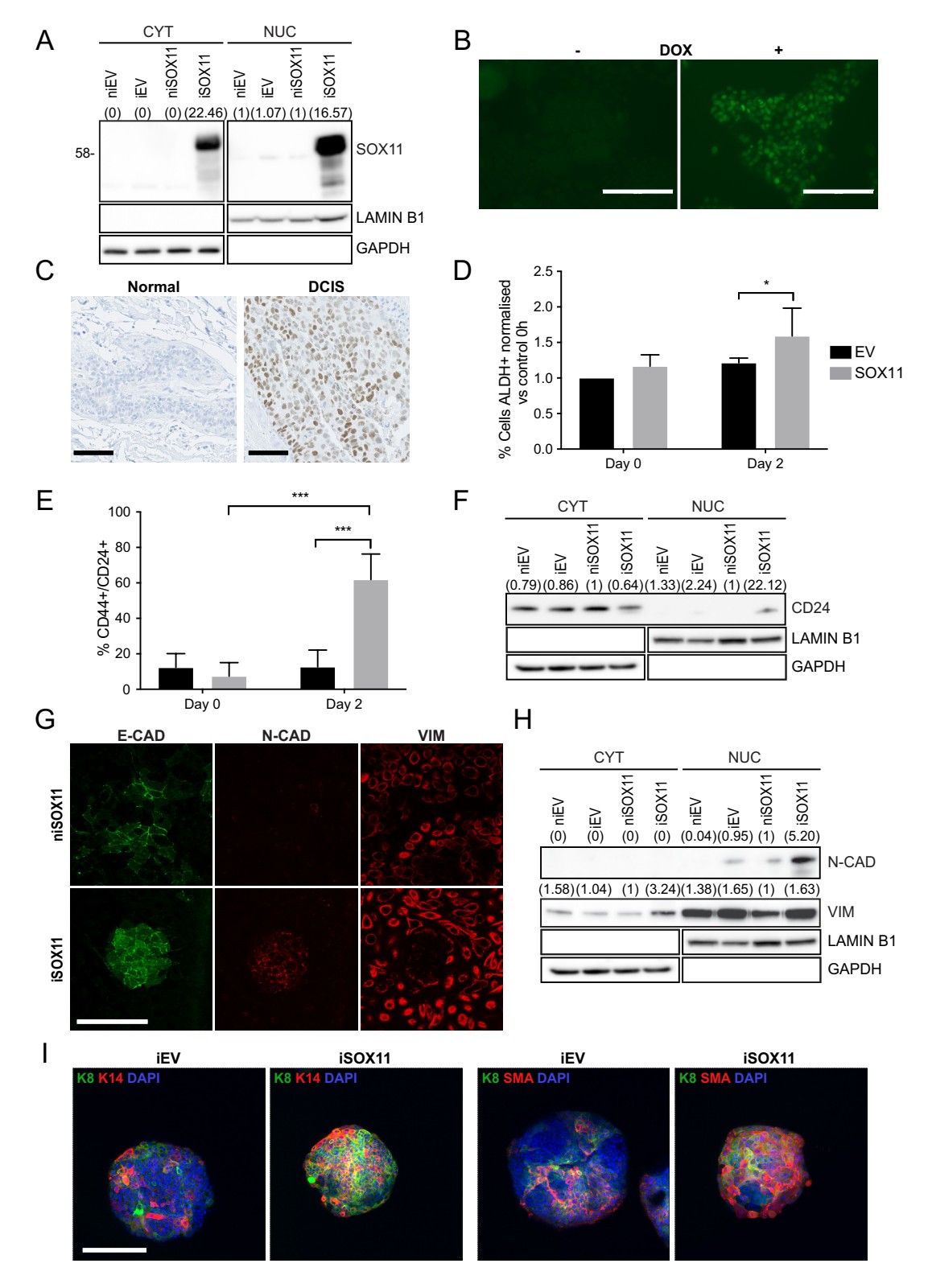

**Figure 1.** Inducible expression of SOX11 leads to changes in cell state profiles of DCIS.com cells. (A) Western blot of SOX11 in cytoplasmic and nuclear fractions of DCIS.com cells containing the pInducer21 empty vector in presence (iEV) or absence (niEV) of 1 µM Doxycycline (DOX) or the pInducer21SOX11 with (iSOX11) or without DOX (niSOX11). GAPDH and LAMIN B1 were used as loading control of cytoplasmic and nuclear fractions, respectively. Densitometry results normalised against niSOX11 are shown in brackets. (B) SOX11 expression detected in iSOX11 cells stained by IF after

*Figure 1 continued on next page*

*Figure 1 continued*

48 hr of DOX induction. Scale Bar: 200 µm. (**C**) ER- DCIS case sample showing SOX11 staining in DCIS and adjacent normal breast tissue. Scale Bar: 200 µm. (**D**) Results from flow cytometry analysis of Aldefluor assays of niEV and niSOX11 cells (day 0) and iEV and iSOX11 after 2 days treatment with 1 µM DOX. Results show the % of ALDH+ cells normalised against niEV. Error bars represent SD. *p = 0.0223. n = 5. (**E**) Results from flow cytometry analysis of CD24 and CD44 of niEV and niSOX11 cells (day 0) and iEV and iSOX11 after treatment with 1 µM DOX for 2 days. Results show the average % of cells CD44+/CD24+ in each condition. Error bars represent SD. ***p = 0.0005 (iSOX11 vs niSOX11) and p = 0.0009 (iSOX11 vs iEV) n = 3. (**F**) Western blot of CD24 in cytoplasmic and nuclear fractions of niEV, niSOX11, iEV and iSOX11 cells. GAPDH and LAMIN B1 were used as loading control of cytoplasmic and nuclear fractions, respectively. In brackets densitometry results normalised against niSOX11. (**G**) Confocal images of IF staining of E-CADHERIN, N-CADHERIN, VIMENTIN in niSOX11 and iSOX11 cells. Cells were grown in slides covered with Collagen I. Scale Bar: 200 µm. (**H**) Western blot of N-CADHERIN and VIMENTIN in cytoplasmic and nuclear fractions of niEV, niSOX11, iEV and iSOX11 cells. GAPDH and LAMIN B1 were used as loading control of cytoplasmic and nuclear fractions, respectively. Densitometry results normalised against niSOX11 are shown in brackets. (**I**) Confocal IF images of iEV and iSOX11 spheroids treated with 1 µM DOX, stained with luminal marker, K8, and basal markers K14 or SMA, and DAPI. Scale bar: 200 µm. DOX: doxycycline, IF: Immunofluorescence.

The online version of this article includes the following figure supplement(s) for figure 1:

**Figure supplement 1.** Inducible expression of SOX11 in DCIS.com cells leads to changes in CD24 profiles.
**Figure supplement 2.** Inducible expression of SOX11 in DCIS.com cells leads to changes in ALDH activity.
**Figure supplement 3.** Inducible expression of SOX11 in DCIS.com cells leads to changes in expression of epithelial and mesenchymal markers.
**Figure supplement 4.** Inducible expression of SOX11 in DCIS.com spheroids leads to changes in epidermal marker expression.
**Figure supplement 5.** Inducible expression of SOX11 in DCIS.com spheroids leads to changes in epidermal marker expression.

proteins attached to the external part of the nuclear envelope since both Lamin B and Vimentin are known to associate with the nuclear envelope (*Figure 1G–H* and *Figure 1—figure supplement 3*; *Georgatos and Blobel, 1987a*; *Georgatos and Blobel, 1987b*). We assessed the expression of markers associated with mammary epithelial and mesenchymal states in iEV and iSOX11 as well as non-induced (denoted ni) niEV and niSOX11 DCIS cells. K5, K8, and VIM levels are elevated in iSOX11 cells compared to iEV cells (*Figure 1—figure supplement 3*). These results are consistent with a role for SOX11 in promoting a mesenchymal state and influencing mammary epithelial phenotypes when expressed in epithelial cells.

To further analyse the effects of SOX11 on mammary epithelial phenotypes, we assessed the expression of markers associated with the two major mammary lineages in DCIS cells grown as spheroids that formed from iSOX11 and to iEV, niEV, and niSOX11 control cells. We detected significant increases in expression of both luminal (K8) and basal (K14, SMA) lineage markers in SOX11-expressing spheroids as well as a significant increase in cells co-expressing markers of both lineages, which is suggestive of embryonic mammary phenotypes (*Figure 1I* and *Figure 1—figure supplements 4–5*).

## DCIS cells expressing SOX11 grow more slowly and form more invasive spheroids

Reduced cell growth was detected in iSOX11 cells compared to control iEV cells grown in both 2D or as spheroids (*Figure 2A–B*). Control iEV cells were more clonogenic than iSOX11 cells whether plated in colony-forming assays or when grown from single cells (*Figure 2—figure supplement 1*). Although iEV cells have greater colony-forming capacity than iSOX11 when large colonies (greater than 50 microns) were counted, a greater number of small colonies (less than 50 microns) formed (*Figure 2—figure supplement 1*). We found DCIS-LacZ control cells also exhibited both greater colony and sphere-forming capacity than DCIS-SOX11 cells (*Figure 2—figure supplement 1*). As with iSOX11 cells, a greater number of small colonies form in colony-formation assays with DCIS-SOX11 cells. Spheroids formed from PKH-labelled DCIS-SOX11 cells are smaller and retain PKH dye more than labelled DCIS-LacZ cells (*Figure 2—figure supplement 1*). These results show that SOX11 expression leads to a reduction of larger spheroids formed from DCIS.com cells and increased formation of smaller spheroids that retain label. Multiple attempts to transduce primary human breast epithelial cells to express SOX11 constitutively with a CMV-driven vector were unsuccessful but are consistent with a possible role for SOX11 in regulating a quiescent state (data not shown).

*SOX11* expression is enriched in ER- and HER2+ invasive breast cancers (*Figure 2—figure supplement 2*). To explore whether sustained reactivation of SOX11 promotes tumour progression, we assessed features of iSOX11 spheroids. Spheroids formed from iSOX11 DCIS cells were smaller and

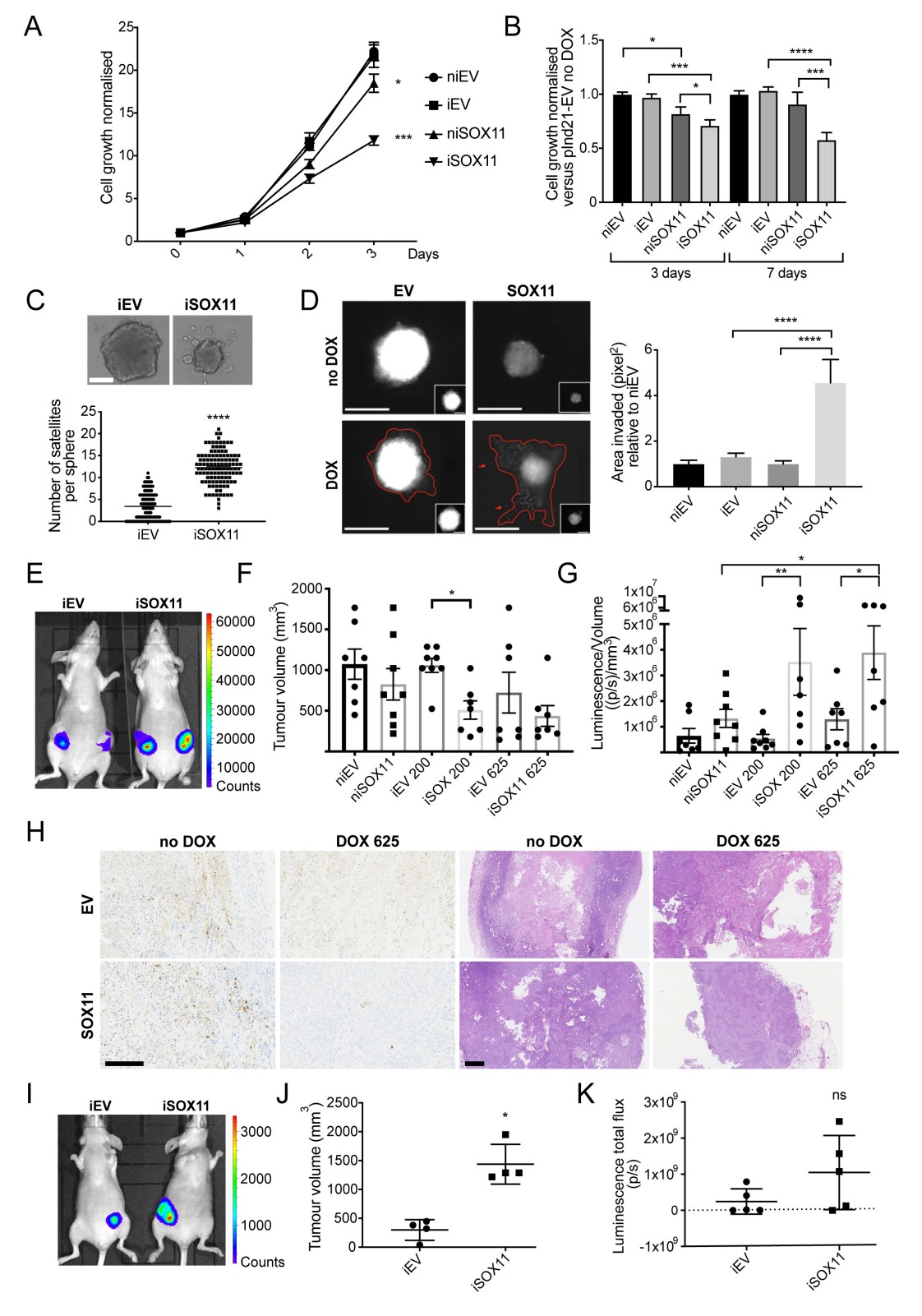

**Figure 2.** DCIS cells expressing SOX11 grow more slowly and form more invasive spheroids. (**A**) Cell growth assays results for iEV or iSOX11 DCIS cells (induced with 1 µM DOX for 72 hr). Experiments were performed five times. Error bars represent SEM. *p = 0.0450 and ****p < 0.0001. (**B**) Cell growth assays results for spheroid formed with iEV or iSOX11 DCIS cells induced with 1 µM DOX for 3 or 7 days. Experiments were performed three times. Error bars represent SEM. p-values (3 days): *p = 0.0374 (niEV vs niSOX11), *p = 0.0221 (niSOX11 vs iSOX11) ***p = 0.0002. p- values (7 days):

*Figure 2 continued on next page*

*Figure 2 continued*

***p = 0.0004 and ****p < 0.0001 (**C**) Examples of DCIS iEV and DCIS iSOX11 spheroids grown on low attachment plates. Graph shows the number of microsatellites per sphere. ****p < 0.0001. (**D**) Invasion assay after overlaying niEV, iEV, niSOX11 and iSOX11 DCIS spheroids with Collagen I. Scale bar: 200 μm. Graph shows the area invaded in pixel$^2$ normalised against niEV. Both ****p < 0.0001. (**E–G**) IVIS imaging, tumour volumes and luminescence total flux/volume results after mammary fat pad xenografts. p-value in F: *p = 0.0331. p-values in G: *p = 0.0252 (niSOX11 vs iSOX11 625ppm), *p = 0.0285 (iEV 625 ppm vs iSOX11 625 ppm), **p = 0.0082. (**H**) Representative images of IHC staining to detect Cleaved Caspase three and H and Es of tumours resected from mice injected with iEV and iSOX11 cells. Scale bar: 500 μm. (**I–K**) IVIS imaging, tumour volumes and luminescence total flux after mammary intraductal injection (MIND) xenografts. *p = 0.0286 (U Mann Whitney). DOX: doxycycline, IHC: Immunohistochemistry.

The online version of this article includes the following figure supplement(s) for figure 2:

**Figure supplement 1.** Inducible expression of SOX11 in DCIS.com cells leads to changes in stem cell activity.

**Figure supplement 2.** *SOX11* levels in breast cancer METABRIC dataset.

**Figure supplement 3.** DCIS cells expressing SOX11 form smaller tumours comprised of more viable cells.

**Figure supplement 4.** Panel shows photomicrographs of primary fat pad tumours following immunohistochemical detection of SOX11, ALDH1A1, CD24, and H and E staining.

formed a higher number of peripheral microspheres than iEV cells (*Figure 2C*). When overlaid with Collagen I, spheroids formed from iSOX11 cells are more invasive compared to iEV, niEV, and niSOX11 control cells (*Figure 2D*). Tumours that formed after mammary fat pad xenografts of iSOX11 cells were smaller than iEV tumours when mice were fed chow with moderate levels (200 or 625 ppm) of DOX (*Figure 2E–F*). Despite their smaller volume, tumours originated from cells with high SOX11 levels displayed greater bioluminescence than control tumours, suggesting iSOX11 tumours contained more viable cells (*Figure 2G*). IHC staining of mammary tumours was performed to detect Cleaved Caspase 3 (CC3). Larger necrotic and CC3+ (apoptotic) areas were observed in the EV tumour tumours and niSOX11 tumours when compared to the iSOX11 tumours, which showed little central necrosis and fewer CC3+ cells (*Figure 2H*). We also observed that extremely small and less luminescent tumours were formed in mice xenografted with iSOX11 cells, compared to control cells that were fed higher levels (1250 or 2000 ppm) of DOX-chow (*Figure 2—figure supplement 3*). Higher levels of CC3 were detected in these iEV tumours, which displayed central necrosis, whilst almost no CC3 or necrosis was detected in iSOX11 tumours (*Figure 2—figure supplement 3*). Tumours formed from iSOX11 DCIS cells grew out quickly when (1250 or 2000 ppm) DOX-chow was replaced with normal chow, suggesting that high levels of SOX11 could keep tumours in a non-proliferative state and that upon DOX withdrawal, proliferation resumed and the ostensible quiescent state is a reversible condition (*Figure 2—figure supplement 3*).

When injected directly into the mammary duct, iSOX11 DCIS cells formed slightly larger tumours, with similar bioluminescence levels as iEV tumours, which indicates the microenvironment highly influences behaviour of SOX11+ tumour cells (*Figure 2I–K*). A greater proportion of mammary tumours formed from iSOX11 cells expressed moderate to high levels of ALDH1 (6/6) compared to control tumours (1/8). In addition, CD24+ cells were observed at greater frequency in iSOX11 tumours compared to control tumours (*Figure 2—figure supplement 4*). Of note, we observed nuclear CD24 staining in iSOX11 tumour cells, but not in iEV, niEV, or niSOX11 tumours, in line with the observed in vitro result (*Figure 2—figure supplement 4*). It has been suggested that cells designated CD24- using flow cytometry maybe expressing CD24 in the nucleus and this promotes aggressive tumour properties, since it has been shown that nuclear CD24 is able to drive tumour growth (*Duex et al., 2017*).

## SOX11 expression promotes expression of developmental pathways frequently activated in cancer

To identify potential SOX11 targets that could confer features of embryonic, epithelial/mesenchymal hybrid state, and metastasis-promoting features to mature breast cells, RNA from iEV, niEV, or niSOX11 control and iSOX11 cells grown in 2D and in 3D (from spheroids at two time-points: 2 days and 5 days after spheroid formation) were sequenced (*Figure 3A*, *Supplementary file 1* and *Figure 3—figure supplement 1*). Uninduced niSOX11 control cells express very low levels of *SOX11*, and suggest the vector is slightly leaky. iSOX11 DCIS cells cultured in 2D showed enrichment of genes regulating actin filament sequestration, phospholipid catabolism, ERBB signaling, chemotaxis, and epithelial differentiation (*Figure 3B* and *Supplementary file 2*). iSOX11 DCIS cells grown in 3D

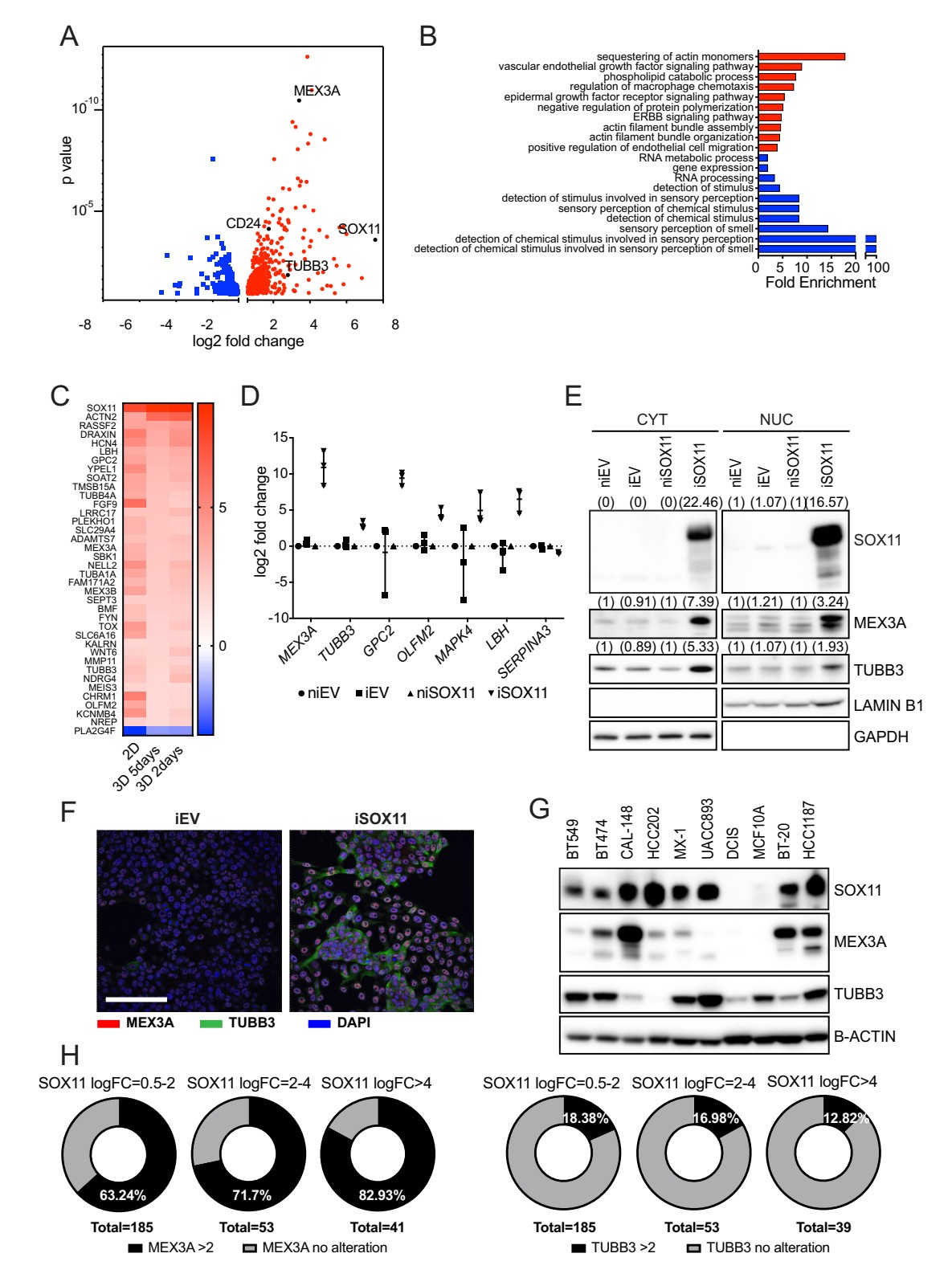

**Figure 3.** SOX11 expression promotes expression of developmental pathways frequently activated in cancer. (**A**) Volcano plot representing the RNAs with a log2 fold-change > +/- 0.585 in the RNA-sequencing results of iSOX11 cells grown in 2D compared with the controls [(iSOX11-niSOX11)- (iEV-niEV)] to account for effects of DOX treatment on DCIS.com cells. (**B**) Gene ontology results from A. (**C**) List of genes overexpressed log2 fold-change > +/-1.585 times in all three RNA-sequencing (cells grown in: 2D, 3D for 2 days, 3D for 5 days) results comparing iSOX11 versus iEV. (**D**) qRT-PCR results

*Figure 3 continued on next page*

*Figure 3 continued*

for several potential SOX11 targets in EV and SOX11 cells with and without DOX induction in cells grown in 2D. Experiment was repeated three times. (E) Western blot of MEX3A and TUBB3 in cytoplasmic and nuclear fractions of EV or SOX11 cells in presence or absence of 1 μM DOX. GAPDH and LAMIN B1 were used as loading control of cytoplasmic and nuclear fractions respectively. In brackets, densitometry results normalised against niEV and niSOX11. (F) IF staining of DCIS iEV and DCIS iSOX11 cells with TUBB3 (green) and MEX3A (red). Scale: 200 μm. (G) Western blot of MEX3A and TUBB3 in SOX11+ breast cancer cell lines and SOX11- DCIS.com and MCF10A from the MCF10A mammary cell progression series. (H) Pie charts representing the percentage of breast cancer samples with a log2 fold-change greater than two in the levels of *MEX3A* or *TUBB3* RNA when *SOX11* increased between 0.5- and 2-fold, 2- and 4-fold, or greater than 4-fold in the TCGA dataset. DOX: doxycycline, qRT-PCR: Quantitative real time PCR, IF: Immunofluorescence.

The online version of this article includes the following figure supplement(s) for figure 3:

**Figure supplement 1.** RNA sequencing results comparing iSOX11 and iEV cells.

showed enrichment of genes regulating ECM disassembly, collagen biosynthesis, glycosaminoglycan metabolism, and platelet degranulation (*Figure 3—figure supplement 1* and *Supplementary file 3*). Platelet activation is one of the first steps of tissue repair as part of the wound healing process and Sox4 and Sox11 have recently been shown to reactivate an embryonic epidermal programme during wound repair in mice (*Miao et al., 2019*). We found substantial overlap of the embryonic wound signature that was shown to be directly regulated by Sox11 and Sox4 (*Miao et al., 2019*) with iSOX11 spheroids (*Table 1*). In particular, we detected upregulation of embryonic wound signature components with links to actin polymerisation and cell adhesion, and one known regulator of embryonic stem cell pluripotency, *RCOR2,* which can function with other transcription factors to induce pluripotent stem cells (*Figure 3—figure supplement 1*; *Yang et al., 2011*).

RNA sequencing analysis detected *CDH2* (encoding N-Cadherin) expressed at significantly higher levels in iSOX11 cells grown in 2D and as spheroids, whilst other EMT markers, including *CDH1* (encoding E-cadherin) and *VIM* were not significantly changed compared to control cells (*Supplementary file 1*). Other notable downstream targets of SOX11 included *MEX3A*, which encodes an RNA-binding protein, that marks slowly proliferating multipotent stem cells in mouse intestine (*Barriga et al., 2017*) and totipotent cells in *C. elegans* (*Pereira et al., 2013*); *MMP11, ST6GALNAC5,* and *TUBB3,* which are highly expressed in breast cancers that metastasize to brain (*Bos et al., 2009*; *Kim et al., 2015*; *Lee et al., 2016*). We confirmed that a number of putative SOX11 targets of interest (*Figure 3C* and *Figure 3—figure supplement 1*, *Supplementary file 2*),

**Table 1.** iSOX11 spheroids express reactivated embryonic wound signature.
Genes upregulated in mouse epidermal cells at E13.5 and at wound edge that are directly regulated by Sox11 and Sox4 in both E16.5 epidermis and keratinocytes in *Miao et al., 2019* are significantly upregulated in iSOX11 cells grown as spheroids.

| 3D (2 days) | Log2 fold-change | p-Values |
|---|---|---|
| GNG2 | 2.30823519 | 4.85E-05 |
| RCOR2 | 1.57478431 | 2.90E-05 |
| MARCKSL1 | 1.06239497 | 7.15E-08 |
| EVL | 0.8482472 | 9.60E-06 |
| SNN | 0.78560135 | 3.11E-05 |
| FBL1M1 | 0.78250288 | 3.12E-05 |
| ETV4 | 0.77064674 | 8.27E-05 |
| VCAN | 0.67955833 | 0.00094939 |
| TWIST2 | 0.64425385 | 0.01949167 |
| PXDN | 0.63841224 | 1.14E-05 |
| ARHGEF2 | 0.58601526 | 2.88E-06 |
| TMSB10 | 0.56607317 | 0.00030783 |
| C4orf48 | 0.54288141 | 0.03406946 |

including *MEX3A* and *TUBB3* were upregulated in iSOX11 cells when measured by qPCR (*Figure 3D*). We also detected upregulation of MEX3A and TUBB3 protein in iSOX11 cells (*Figure 3E–F*). We found many SOX11+ breast cancer cell lines express high levels of *MEX3A* or *TUBB3* compared to DCIS.com cell line (*Figure 3G*). *MEX3A* and *TUBB3* levels are correlated with *SOX11* expression in breast cancers in the TCGA dataset (*Figure 3H*) and in breast cancer cell lines in the Broad dataset (*Ghandi et al., 2019*; *Figure 3—figure supplement 1*). Notably, with greater increases of *SOX11* levels, a higher percentage of samples with increased *MEX3A* are observed. These findings support a role for SOX11 in mediating developmental signals during breast cancer progression.

## DCIS cells expressing SOX11 show alterations in metastatic tropism

To explore whether sustained reactivation of SOX11 promotes tumour progression, we injected luciferase-tagged iEV and iSOX11 cells into the mammary fat pad. Four weeks after orthotopic xenografting with iSOX11 cells, brain micrometastases were detected in two out of six mice by IVIS imaging, whilst none were observed in six mice xenografted with control cells (*Figure 4A*). Liver and lung micrometastases were detected in both cohorts when assessed by IVIS imaging.

After injection into the tail vein, IVIS imaging detected tumour cells in lungs in seven out of the eight mice, bone in three out of eight mice, and in brain in one of eight mice xenografted with iSOX11 cells, whilst no bone or brain metastases were observed in eight mice xenografted with control cells (*Figure 4B*). It was noted that the frequencies of mice with lung metastasis were similar, but a significant reduction of iSOX11 DCIS cells accumulated in the lungs was observed when compared to mice engrafted with iEV cells when quantified by IVIS (*Figure 4C–D*). Macroscopic examination of metastatic lesions confirmed the reduction in tumour burden (data not shown).

*SOX11* expression in primary breast cancer is associated with increased metastasis formation at distant sites (*Figure 4E*). *SOX11* is highly expressed in brain metastases and is also detected in bone metastasis from breast cancer patients (*Figure 4—figure supplement 1*). *SOX11* amplification and overexpression has been detected in breast cancer brain metastasis in patients with ER-, ER+, and *BRCA1-/-* tumours in a recent study that comprehensively profiled a small number of cases (*Figure 4—figure supplement 1*; *Saunus et al., 2015*). In another dataset in which samples from 21 breast cancer brain metastasis (BCBM) patients were transcriptionally profiled by RNA sequencing, elevated levels of *SOX11* are detected in the brain metastasis in one third of cases compared to the primary tumour (*Figure 4—figure supplement 2*; *Varešlija et al., 2019*).

SOX11+ DCIS.com cells from brain metastasis display a colonisation and growth advantage after intracranial xenografting iSOX11 cells were isolated from mouse brain after orthotopic injections and expanded (designated iSOX11Br). Induction of SOX11 with DOX led to increased expression of SOX11, MEX3A and TUBB3 compared to untreated cells (*Figure 5A*). Other cells isolated and expanded from bone and lung metastasis after orthotopic injections of iSOX11 did not express SOX11 after induction with DOX.

After xenografting iSOX11Br cells into the tail vein, mice that had been fed normal chow had a greater metastatic burden in the lungs compared to mice fed DOX-containing chow to induce SOX11 expression, similar to results observed using the parental iSOX11 DCIS cell line (*Figure 5B–D*). After intracranial injections of iSOX11Br cells, higher levels of bioluminescence were detected in the brains of mice fed DOX chow (*Figure 5E*). Induction of SOX11 expression led to larger tumour burden in the brain and reduced survival (*Figure 5F*). These results indicate that iSOX11Br cells have a colonisation and growth advantage in the brain compared to that niSOX11Br cells lacking SOX11 expression.

## SOX11 regulates proliferative state of ER- breast cancer cells

Next, we examined the effect of reducing SOX11 levels in CAL-148 ER- breast cancer cell line that expresses very high SOX11 levels. Using DOX-inducible lentiviral vectors containing shRNAs to *SOX11* and control non-targeting shRNAs, reductions in *SOX11* levels were obtained, as well as decreases of *TUBB3, MEX3A, GPC2, MPK4, OLFM2, ST6GALNAC5,* and *NCAD* levels and an increase in *SERPINA3* in CAL-148 cells after *SOX11* knockdown when compared to control (*Figure 6A*). Reduced levels of SOX11, MEX3A and TUBB3 protein were detected after *SOX11* knockdown (*Figure 6B*). Cell viability assays detected greater cell numbers after *SOX11* knockdown

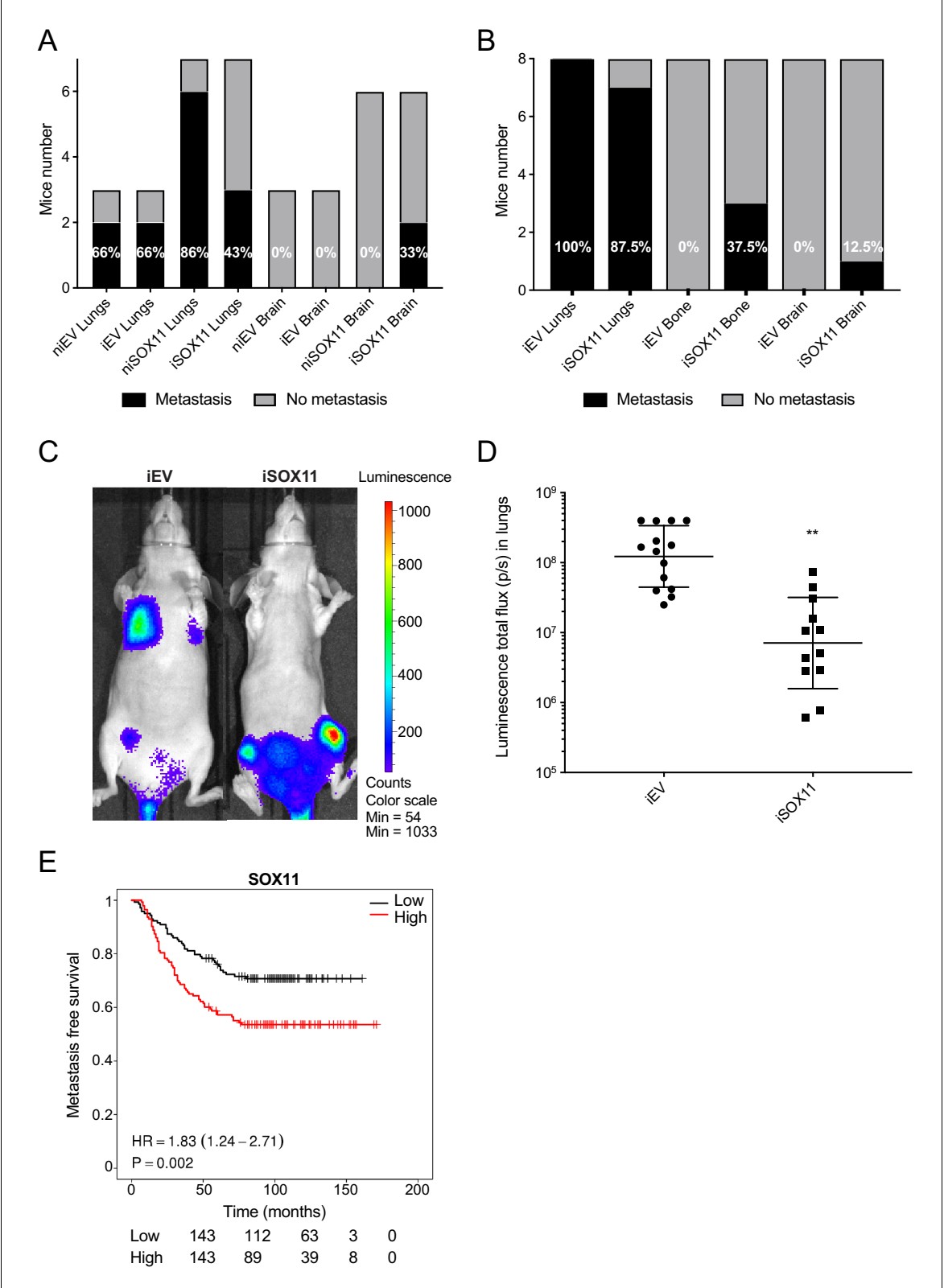

**Figure 4.** DCIS cells expressing SOX11 show alterations in metastatic tropism. (**A**) Tabulated results of micrometastasis assessed by ex vivo IVIS imaging after orthotopic mammary fat pad xenografting of iEV and iSOX11 cells. (**B**) Tabulated results of micrometastasis assessed by ex vivo IVIS imaging after xenografting iEV and iSOX11 cells via the tail vein. (**C**) Representative in vivo IVIS 7 days after tail vein injections of iEV and iSOX11 cells. *Figure 4 continued on next page*

Figure 4 continued

(D) Quantification of in vivo lung metastatic burden at day 31. Graph shows the luminescence total flux (p/s) in the lungs 31 days after tail vein injections. **p = 0.0011. (E) Distance metastasis-free survival curve for *SOX11* in breast cancer patients in the Wang cohort (GSE2034).

The online version of this article includes the following figure supplement(s) for figure 4:

**Figure supplement 1.** *SOX11*expression in breast cancer metastasis.
**Figure supplement 2.** *SOX11*expression in breast cancer brain metastasis.

in CAL-148 cells, compared to control cells (*Figure 6C*). Colony-formation assays detected an increase in clonogenetic potential of CAL-148 cells with reduced *SOX11* levels (*Figure 6D*). These results suggest SOX11 could regulate proliferative state of stem cells in ER- breast cancer cells.

## TUBB3, an established SOX11 target, regulates proliferation and invasion of ER- breast cancer cells

As with *SOX11*, both distant metastasis-free survival and overall survival of breast cancer patients are reduced when high levels of *TUBB3,* an established SOX11 target in neural cells, are expressed in primary tumours (*Figure 7—figure supplement 1*). We next examined the effects of TUBB3 on triple negative breast cancer (TNBC) growth and invasion. Using siRNA-mediated knockdown, we found that reducing TUBB3 levels in BT-20 cells resulted in reduced growth in both 2D culture and spheroid culture (*Figure 7A–C*). More cells were arrested in G2/M phase of the cell cycle when TUBB3 levels were reduced (*Figure 7D* and *Figure 7—figure supplement 2*), consistent with a known role of TUBB3 in regulating cell cycle progression of tumour cells. As a result of cell cycle arrest, an increase of multinucleated cells in the >G2/M phase and a higher proportion of dead cells in sub-G1 phase were detected. Spheroid invasion assay s detected less invasion when TUBB3 levels were reduced (*Figure 7E–F*). These results suggest SOX11 can regulate proliferation and invasive growth through TUBB3 in ER- breast cancer cells.

## MEX3A, a novel potential SOX11 downstream effector, regulates cell growth and E/M state of ER- breast cancer cells

Both distant metastasis-free survival and overall survival of breast cancer patients are reduced when high levels of *MEX3A* are expressed in primary tumours, as would be expected for a SOX11 target (*Figure 7—figure supplement 1*). Due to MEX3A's established links with regulation of both EMT and proliferation of intestinal and various types of cancer cells, we knocked down *MEX3A* in ER- BT-20, CAL-148 and HCC1187 breast cancer cells to determine if MEX3A regulates similar processes. In HCC1187 and BT-20 cells, a slight but significant reduction of invasive growth was detected when MEX3A levels were reduced with some, but not all siRNAs (*Figure 8A–B* and *Figure 8—figure supplement 1*). An increase in cell numbers were observed after knockdown of *MEX3A* in both CAL-148 and HCC1187 cells grown in 2D or 3D (*Figure 8C–D*). Cell cycle analysis detected a reduction in S phase after *MEX3A* knockdown in both cell lines (*Figure 8E*, *Figure 8—figure supplement 1*). Several candidate cell cycle regulators from a consensus stem cell quiescence signature (*Cheung and Rando, 2013*) were downregulated in iSOX11 cell signatures, including *RRM2* and *SURVIVIN* (*Figure 8—figure supplement 1* and *Supplementary file 3*). Western blotting found that both RRM2 and SURVIVIN were upregulated when MEX3A levels were reduced in CAL-148 and BT-20 cells (*Figure 8F* and *Figure 8—figure supplement 1*). High expression levels of *MEX3A* co-occur with increased levels of *E2F3*, *CCNE1* and *CDKN2A,* and decreases in *RB1* levels in the TCGA dataset (*Figure 8—figure supplement 1*); *MEX3A* levels show strong correlation with the levels of *E2F3* and *CCNE1* (*Supplementary file 4*). The addition of EGF to CAL-148 cells growing in either serum-free or low-serum media led to reduction of MEX3A levels (*Figure 8—figure supplement 2*). CAL-148 cells normally form aggregates when grown in 2D culture conditions and after *MEX3A* knockdown, CAL-148 cells displayed a reduced ability to form aggregates and an acquired ability to adhere to plastic (*Figure 8G*). After MEX3A knockdown, CAL-148 cells displayed increased expression of E-CADHERIN and EPCAM (*Figure 8H*). BT-20 and HCC1187 cells lack expression of E-CADHERIN but showed an increase in EPCAM levels after MEX3A knockdown (*Figure 8—figure supplement 2*). Together, these results are consistent with roles for MEX3A in regulation of cell growth and EMT in SOX11+ ER- breast cancer cells.

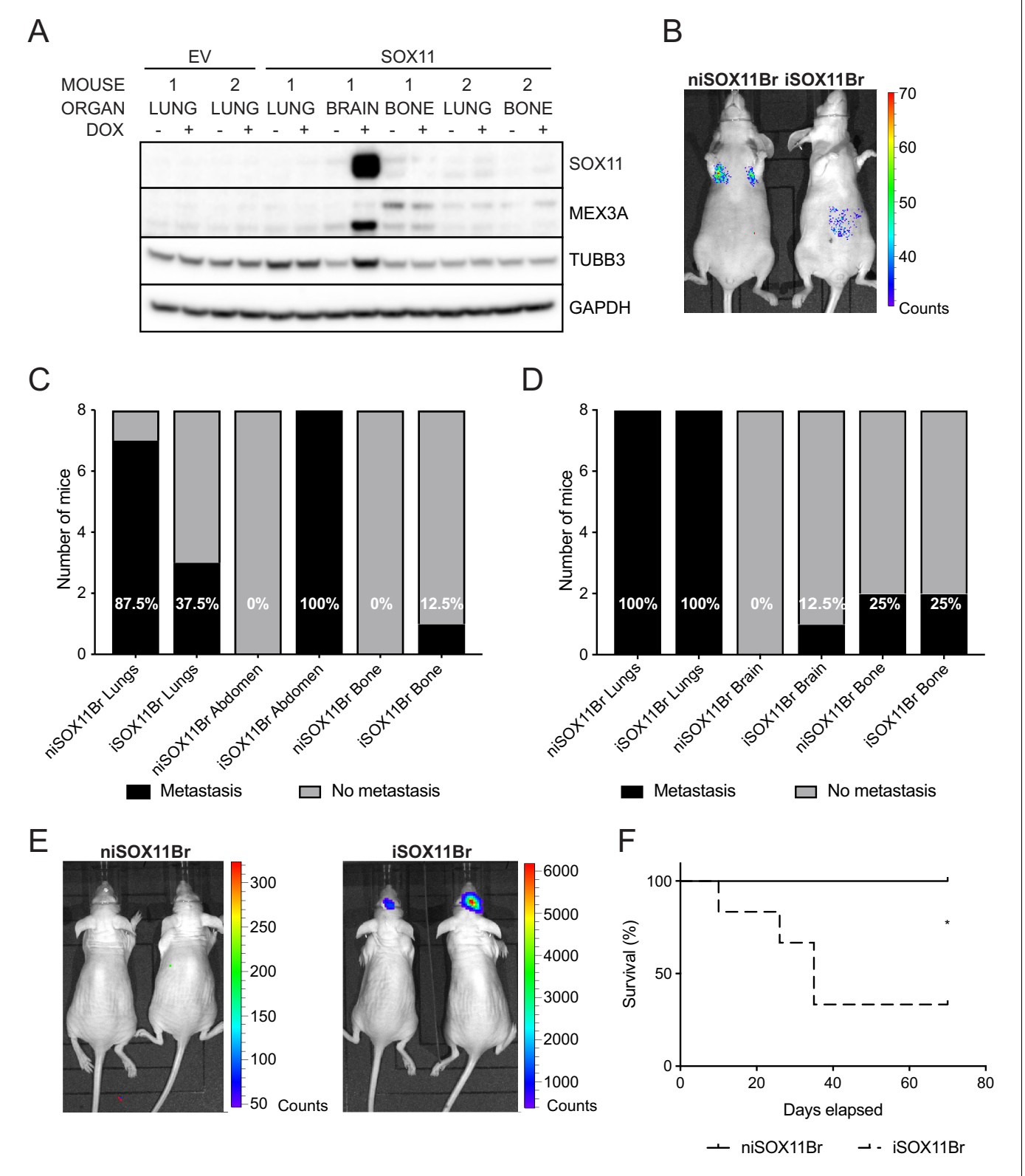

**Figure 5.** SOX11+ DCIS cells isolated from brain metastasis display a colonisation and growth advantage after intracranial xenografting. (A) Western blot of SOX11, MEX3A and TUBB3 in total cell lysates of EV and SOX11 cell lines isolated from primary metastasis at indicated sites in presence or absence of DOX. (B) Representative in vivo IVIS imaging 7 days after tail vein injections of iSOX11 cells that were isolated from the brain metastasis (SOX11Br) in presence or absence of DOX. (C) Tabulated results of micrometastasis from in vivo IVIS imaging 7 days after tail vein injections of SOX11Br

*Figure 5 continued on next page*

Figure 5 continued

cells. (D) Tabulated results of micrometastasis from ex vivo IVIS imaging of the tail vein injections of SOX11Br cells. (E) IVIS imaging of mice fed normal chow or DOX-containing chow 10 days after intracranial injections of SOX11Br cells. (F) Survival curve for mice shown in E. *p = 0.0195. DOX: doxycycline.

## Discussion

We previously showed that SOX11 promotes invasive transition of DCIS cells using both in vitro and in vivo models where SOX11 is expressed in DCIS.com cells under the control of a constitutive CMV-driven promoter (*Oliemuller et al., 2017*). Here, we investigated the role of SOX11 in breast cancer progression after tumour formation using a doxycycline-inducible EF1A promoter to express SOX11 in DCIS.com cells at a level comparable to that observed in clinical DCIS and breast cancer samples. DCIS cells induced to express SOX11 prior to spheroid formation form smaller spheroids, which display more invasion compared to control spheroids, recapitulating the phenotypes we observed with constitutive expression of SOX11 (*Oliemuller et al., 2017*). In addition, we observed unique phenotypes using these experimental conditions, including substantial morphological changes (cell detachment and formation of multiple satellite spheroids indicative of a highly invasive phenotype without hydrogel), suggesting breast lesions expressing very high levels of SOX11 possess an inherent high potential to form metastatic lesions.

Stem cells and progenitors may serve as a cell of origin for breast cancers. Many SOX (SRY-related HMG-box) transcription factors are expressed in the postnatal breast and some have been shown to control normal and/or CSCs (*Domenici et al., 2019*; *Kogata et al., 2018*; *Mehta et al., 2019*). SOX11 is a unique transcription factor, since it does not appear to be expressed in normal mammary epithelial cells after birth in either mouse or human and therefore is well-poised to reactivate developmental pathways when expressed in breast cancers (*Oliemuller et al., 2017*; *Tsang et al., 2020*; *Wansbury et al., 2011*; *Zvelebil et al., 2013*).

CD24 and ALDH1 are widely used CSC markers in breast cancer (*Liu et al., 2014*). It is now widely accepted that stem cell states fluctuate and are not fixed (*Liu et al., 2014*) and cells transition between the two states (*Liu et al., 2014*; *O'Brien-Ball and Biddle, 2017*). Our results suggest a dose-dependent effect of SOX11 on CD24 levels. It is possible that CMV-driven versus EF1A-driven SOX11 has distinct cell context effects. Lower levels of CMV-driven SOX11 expression results in expansion of the ALDH+/CD24- CSC population in DCIS.com cells (*Oliemuller et al., 2017*). Meanwhile, EF1A-driven inducible SOX11 expression, presented here, leads to expansion of the CD24+ CSC population and higher levels of SOX11 are associated with an increased size of the ALDH+/CD24+ CSC population. Triple negative breast cancer (TNBC) cells exhibit robust expression of *CD24*, suggesting that the inducible model described here mimics what is observed in human TNBC and may present a potential therapeutic opportunity for some SOX11+ breast cancers (*Barkal et al., 2019*).

*Sox11* is highly expressed in prenatal mammary epithelial cells from the time the mammary organ initially forms (*Zvelebil et al., 2013*). scRNA-sequencing also detects *Sox11* expression by the majority of Lgr5+ embryonic mammary epithelial cells isolated at E14.5, a stage when embryonic mammary epithelial cells are multipotent and most cells express markers associated with the two major mammary lineages (the basal/myoepithelial and luminal lineages) (*Lilja et al., 2018*; *Wuidart et al., 2018*). Prenatal human breast tissues also express markers associated with both basal and luminal mammary lineages (*Jolicoeur et al., 2003*). Co-expression of basal and luminal markers are observed in iSOX11 cells and spheroids formed from them. Our findings are consistent with SOX11 marking populations enriched with embryonic mammary cell phenotypes (*Bland and Howard, 2018*; *Wuidart et al., 2018*).

Epithelial cells with mesenchymal features have been detected at high frequency during organogenesis in the developing mouse embryo using single cell RNA-seq analysis. *Sox11* positively regulates expression of mesenchymal markers, including *N-cadherin (Cdh2)* and *Fibronectin1 (Fn1)* in several developing organs, including intestine, liver, lung, and skin (*Dong et al., 2018*; *Halbleib and Nelson, 2006*). A significant association of *SOX11* and *N-CADHERIN* (*CDH2*) expression is also observed in genomic analyses across six cancer types, suggesting that mesenchymal pathways may be activated by SOX11 in both normal and cancer cells (*Vervoort et al., 2013*). SOX11 has

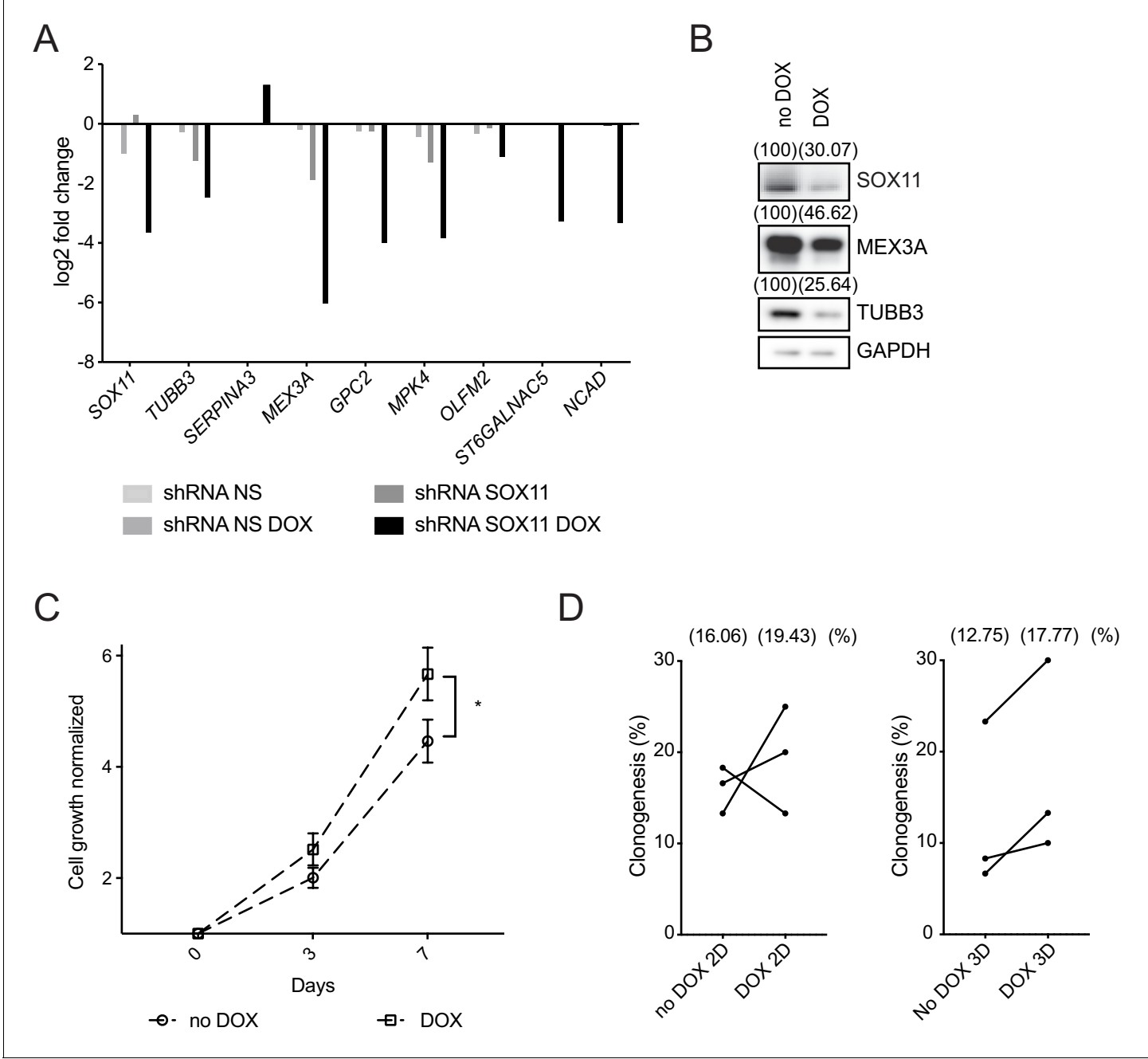

**Figure 6.** SOX11 regulates growth of ER- breast cancer cells. (**A**) qRT-PCR results for several potential SOX11 targets in CAL-148 cells transduced with shRNA to *SOX11* or shRNA NS cells with and without DOX induction in cells grown in 2D. (**B**) Western blot of SOX11, MEX3A and TUBB3 in total cell lysates of CAL-148 cells transduced with shRNA *SOX11* in presence or absence of 1 μM DOX after 48 hr. GAPDH was used as loading control. Densitometry results normalised against no DOX are shown in brackets. (**C**) Cell growth assay results for CAL-148 shRNA *SOX11* cells induced with 1 μM DOX at 3 and 7 days. Experiments were performed three times. Error bars represent SEM. *p=0.0106 (day 7). (**D**) Quantification of clonogenicity in 2D and 3D from single CAL-148 shRNA *SOX11* cells plated in presence or absence of DOX after 21 days. The number in brackets represents the mean in each group of the three experimental replicates. DOX: doxycycline, NS: non-silencing.

previously been implicated in participating in the regulation of epithelial-mesenchymal transition (EMT) (*Venkov et al., 2011*). EMT, stemness and plasticity are intertwined (*Nieto et al., 2016*; *Wahl and Spike, 2017*) and SOX11 is well poised to function as a key regulator of epithelial/mesenchymal cell states during development and in cancers.

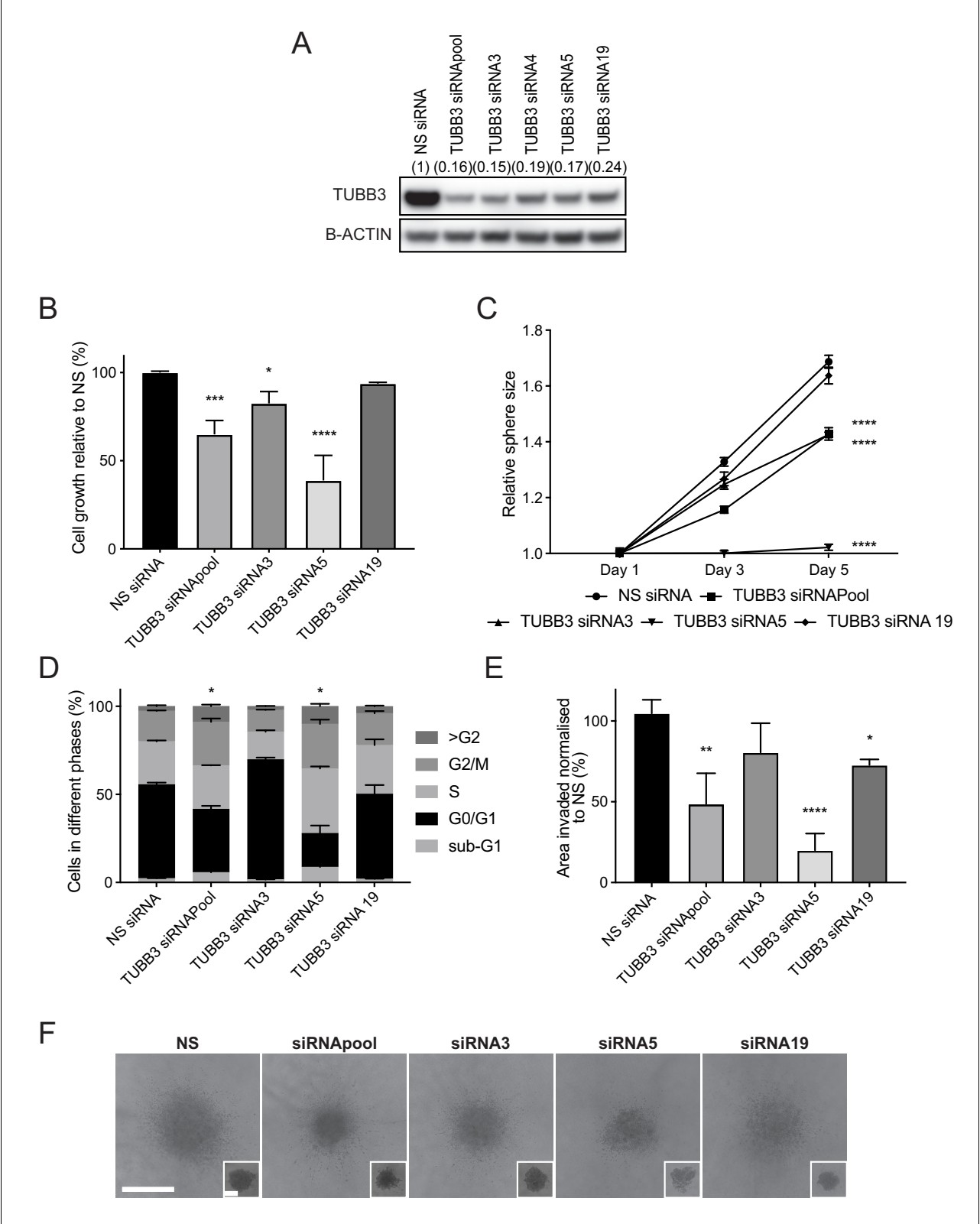

**Figure 7.** TUBB3, an established SOX11 target, regulates cell growth and invasive potential of ER- breast cancer cells. (**A**) Western blot of TUBB3 in total cell lysates of BT-20 cells transfected with siRNAs specific for *TUBB3*. B-ACTIN was used as loading control. Densitometry results normalised against NS siRNA are shown in brackets. (**B**) Cell growth assay results 5 days after BT-20 cells were transfected with siRNA specific for *TUBB3* (since siRNAs require 48 hr for efficient knockdown). Results relative to NS siRNA are shown. Experiments performed three times. *p = 0.0297, ***p = 0.0001,
*Figure 7 continued on next page*

*Figure 7 continued*

****p < 0.0001. (**C**) Sphere size measured 3 and 5 days after BT-20 cells were transfected with siRNA specific for *TUBB3*. Results relative to NS siRNA are shown. Experiments performed three times. ****p < 0.0001. (**D**) Cell cycle analysis performed by flow cytometry at day two after siRNA transfection specific for *TUBB3* in BT-20 cells. Graph shows % of cell in each phase of the cell cycle. *p = 0.0396 (Phase G2/M siRNApool vs siRNA NS) and *p = 0.0243 (Phase G2/M siRNA5 vs siRNA NS). Experiments were performed three times. (**E**) Invasion assay after overlaying BT-20 spheroids with Collagen I at day 2 after siRNA transfection specific for *TUBB3*. Graph shows the area invaded in pixel$^2$ normalised against NS siRNA. *p = 0.0444, **p = 0.0014, ****p < 0.0001. (**F**) Representative images of BT-20 spheroids transfected with indicated siRNAs to *TUBB3* 48 hr after adding Collagen I and (in small insets) at time 0 hr (2 days after transfection). Scale bar: 200 μm. NS: non-silencing.

The online version of this article includes the following figure supplement(s) for figure 7:

**Figure supplement 1.** Survival curves of breast cancer patients with tumours expressing low or high levels of*CD24*,*TUBB3*, and*MEX3A*.
**Figure supplement 2.** Representative flow cytometry histograms of cell cycle analysis of BT-20 cells after transfection with specific *TUBB3* siRNAs.

The regulation of N-CADHERIN (CDH2) by SOX11 is significant since CDH2 promotes motility in human breast cancer cells regardless of their E-CADHERIN expression (*Nieman et al., 1999*). CDH2 is a predictive biomarker for distant metastasis in early-stage breast cancer (*Aleskandarany et al., 2015*), that is commonly detected in breast cancer cells and provides a mechanism for transendothelial migration (*Qi et al., 2005*).

Micrometastasis to brain is detected in ~30% of mice within 1 month of xenografting DCIS cells induced to express high levels of SOX11. *SOX11* is amplified and overexpressed in ~30% of brain metastases in a recent study of a small cohort of breast cancer patients that were profiled using an integrated genomic and transcriptomic analysis of fresh frozen tumour samples (*Saunus et al., 2017*; *Saunus et al., 2015*) and highly expressed by ~30% of BCBM in another independent study (*Vareslija et al., 2019*). Our finding that iSOX11 mammary tumours spontaneously metastasize to the brain is clinically relevant.

We have identified a number of genes that are regulated by SOX11 in our mouse models of breast cancer that are also associated with SOX11 expression in both primary breast cancer and breast cancer brain metastasis. A number of these, such as *TUBB3*, encode targetable molecules that are being used in novel pharmacogenetic approaches in combination with other factors (*Karki et al., 2013*). Others, such as MEX3A, may provide novel biomarkers or therapeutic targets (*Bufalieri et al., 2020*; *Wang et al., 2020*; *Yang et al., 2020*). MEX3A controls the polarity and stemness and affects the cell cycle of intestinal epithelial cells through the downregulation of the mRNA encoding the CDX2 transcription factor (*Pereira et al., 2013*) and, in addition, exhibits a transforming activity when overexpressed in gastric epithelial cells (*Jiang et al., 2012*). The intestine has a reservoir of quiescent stem cells that are resistant to chemotherapy that are marked by Mex3a and are multipotent so have the capacity to produce any kind of intestinal cell and contribute to tumour heterogeneity (*Barriga et al., 2017*). MEX3A may mark a rare mammary stem cell in the human breast that could escape traditional chemotherapy treatments, but this remains to be demonstrated.

The analysis of co-occurrence of *SOX11* with *MEX3A* and *TUBB3* in breast cancers profiled in TCGA suggests that *MEX3A* is expressed predominantly in samples with higher levels of *SOX11*. The differences in cell growth observed both in vitro and in vivo with increasing levels of SOX11 suggest a dose-dependent regulation of SOX11 downstream targets and/or a possible post-translational modification. Also, the discrepancies between the tumour size observed after xenografting iSOX11 cells via the mammary fat pad model and the MIND model, as well as the finding that EGF can decrease MEX3A levels, suggest an influence of microenvironmental factors on SOX11 tumour cell behaviour. This is in agreement with differential expression of *TUBB3* and *MEX3A*, depending if DCIS-SOX11 cells were xenografted in the mammary fat pad or via MIND (*Oliemuller et al., 2017*). Together, this data suggests it may be necessary to classify SOX11+ tumours, depending on SOX11 expression level, as well as the expression of its effectors in order to stratify breast cancer patients. The subcellular localisation of SOX11 can be informative for cancer classification since high SOX11 mRNA levels and detection of the nuclear protein are reliable markers of mantle cell lymphoma (*Mozos et al., 2009*). During neurogenesis, SOX11 is detected in both the nucleus and cytoplasm (*Balta et al., 2018*). Phosphorylation of specific serine residues can modulate SOX11 subcellular localisation and prevent its nuclear localisation (*Balta et al., 2018*). We have not determined

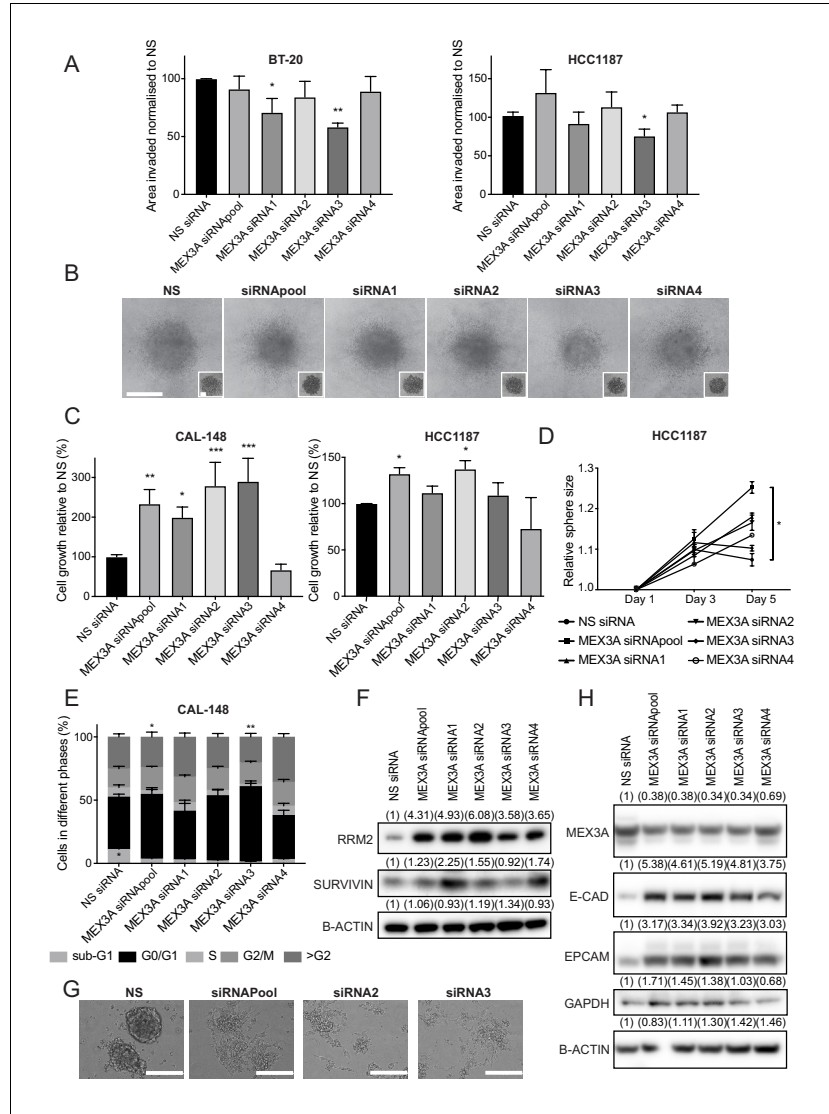

**Figure 8.** MEX3A, a novel SOX11 downstream effector, regulates cell growth and (E/M) state of ER- breast cancer cells. (A) Invasion assay results after overlaying BT-20 and HCC1187 spheroids with Collagen I, at day two after siRNA transfection specific for *MEX3A* or control (NS). Graph shows the area invaded in pixel$^2$ normalised against NS siRNA. *p = 0.0181, **p = 0.0014 for BT-20 and *p = 0.0220 for HCC1187 cells. (B) Representative images of BT-20 spheroids 48 hr after adding Collagen I and at time 0 hr (shown in small inset, 2 days after transfection) made from cells transfected with either control siRNA or *MEX3A* siRNAs. Scale bar: 200 μm. (C) Cell growth assays 5 days after CAL-148 and HCC1187 cells were transfected with siRNA specific for *MEX3A* or NS controls (siRNAs require 48 hr for efficient knockdown; this was taken into account to select day five as final point). Relative results to NS siRNA are shown. Experiments performed three times. *p = 0.0342, **p = 0.0052 ***p = 0.0005 (siRNA2), ****p < 0.0003 (siRNA3) for CAL-148 cells and *p=0.0337 (siRNApool) and *p = 0.0477 for HCC1187 cells. (D) Sphere size measured 3, and 5 days after HCC1187 cells were transfected with siRNA specific for *MEX3A*. Relative results to NS siRNA are shown. Experiments performed three times *p = 0.0240. (E) Cell cycle analysis performed by flow cytometry at day 4 after siRNA transfection specific for *MEX3A* in CAL-148 cells. Graph shows % of cell in each phase of the cell cycle. *p = 0.0147 (phase S siRNApool vs siRNA NS) and *p = 0.0094 (phase S siRNA3 vs siRNA NS) and *p = 0.0419 (phase subG1 NS vs siRNA3). Experiments performed four times. (F) Western blot of RRM2 and SURVIVIN in total cell lysates of CAL-148 at day four after siRNA transfection specific for *MEX3A*. β-ACTIN was used as loading control. Densitometry results normalised against NS siRNA are shown in brackets. (G) Examples of morphological changes observed in CAL-148 cells after 4 days of *MEX3A* knockdown compared to NS control. Scale bar: 200 μm. (H) Western Blot of EPCAM and E-CADHERIN in CAL-148 cells transfected with siRNA specific for *MEX3A* or NS control. β-ACTIN was used as loading control. Densitometry results normalised against NS siRNA are in brackets. NS: non-silencing.

*Figure 8 continued on next page*

*Figure 8 continued*

The online version of this article includes the following figure supplement(s) for figure 8:

**Figure supplement 1.** MEX3A regulates growth of ER- breast cancer cells.

**Figure supplement 2.** MEX3A regulates epidermal state of ER- breast cancer cells.

whether SOX11 localises to the cytoplasm, as well as the nucleus, in breast cancer case samples, and if SOX11 localisation could be useful for classification of breast cancers.

SOX11+ breast cancer cells express markers indicative of phenotypic plasticity and have a high tendency to undergo metastasis (*O'Brien-Ball and Biddle, 2017*). Together, these data suggest that patients whose DCIS and primary breast cancers express high levels of *SOX11* are among a high-risk metastasis subgroup that should be considered for aggressive therapies in neo-adjuvant settings.

# Materials and methods

**Key resources table**

| Reagent type (species) or resource | Designation | Source or reference | Identifiers | Additional information |
|---|---|---|---|---|
| Gene (*Homo sapiens*) | *SOX11* | DNASU | Gene ID: 6664 Clone HsCD00295480 19 | |
| Strain, strain background (*Mus musculus*, Female) | NSG-Foxn1$^{null}$ | in house | | from breeding colony at ICR Biological Services Unit |
| Cell line (*Homo sapiens*) | MCF10DCIS.com | Dr. Gillian Farnie | | |
| Cell line (*Homo sapiens*) | CAL-148 | DMSZ | ACC 460 | |
| Cell line (*Homo sapiens*) | BT-20 | ATCC | HTB-19 | |
| Cell line (*Homo sapiens*) | HCC1187 | ATCC | CRL-2322 | |
| Cell line (*Homo sapiens*) | MCF10A | ATCC | CRL-10317 | |
| Cell line (*Homo sapiens*) | BT474 | ATCC | HTB-20 | |
| Cell line (*Homo sapiens*) | BT549 | ATCC | HTB-122 | |
| Cell line (*Homo sapiens*) | HCC202 | Dr. Paul Huang, ICR, London | CRL-2316 | |
| Cell line (*Homo sapiens*) | MX-1 | DMSZ | CVCL_4774 | |
| Cell line (*Homo sapiens*) | UACC893 | ATCC | CRL-1902 | |
| Transfected construct (*Homo sapiens*) | pInducer21-SOX11 | this paper | SOX11 coding sequence (GENEID: 6664) was subcloned into pInducer21 (ORF-EG) plasmid Cat# 46948 (See Expression vectors in Materials and methods) | Lentiviral construct to transfect and express SOX11 sequence |
| Transfected construct (*Homo sapiens*) | pInducer13-SOX11 shRNA 174 | this paper | specific shRNA174 for SOX11 from pGIPZ plasmids from Horizon was subcloned into pInducer13 (miR-LUP) plasmid Cat# 46936 (See Expression vectors in Materials and methods) | Lentiviral construct to transfect and express the shRNA |
| Transfected construct (*Homo sapiens*) | pInducer13-shRNA NS | this paper | NS: non-silencing shRNA from pGIPZ plasmid from Horizon was subcloned into pInducer13 (miR-LUP) plasmid Cat# 46936 (See Expression vectors in Materials and methods) | Lentiviral construct to transfect and express the shRNA |

*Continued on next page*

*Continued*

| Reagent type (species) or resource | Designation | Source or reference | Identifiers | Additional information |
|---|---|---|---|---|
| Transfected construct (*Homo sapiens*) | siRNA: siGENOME Non-Targeting siRNA #1 | Horizon Discovery | D-001210-01-20 | |
| Transfected construct (*Homo sapiens*) | siRNA: siGENOME SMARTpool TUBB3 siRNA | Horizon Discovery | MQ-020099-03-0020 | |
| Transfected construct (*Homo sapiens*) | siRNA: siGENOME TUBB3 siRNA #3 | Horizon Discovery | MQ-020099-03-0020 | |
| Transfected construct (*Homo sapiens*) | siRNA: siGENOME TUBB3 siRNA #4 | Horizon Discovery | MQ-020099-03-0020 | |
| Transfected construct (*Homo sapiens*) | siRNA: siGENOME TUBB3 siRNA #5 | Horizon Discovery | MQ-020099-03-0020 | |
| Transfected construct (*Homo sapiens*) | siRNA: siGENOME TUBB3 siRNA #19 | Horizon Discovery | MQ-020099-03-0020 | |
| Transfected construct (*Homo sapiens*) | siRNA: siGENOME SMARTpool MEX3A siRNA | Horizon Discovery | MQ-022355-01-0020 | |
| Transfected construct (*Homo sapiens*) | siRNA: siGENOME MEX3A siRNA #1 | Horizon Discovery | MQ-022355-01-0020 | |
| Transfected construct (*Homo sapiens*) | siRNA: siGENOME MEX3A siRNA #2 | Horizon Discovery | MQ-022355-01-0020 | |
| Transfected construct (*Homo sapiens*) | siRNA: siGENOME MEX3A siRNA #3 | Horizon Discovery | MQ-022355-01-0020 | |
| Transfected construct (*Homo sapiens*) | siRNA: siGENOME MEX3A siRNA #4 | Horizon Discovery | MQ-022355-01-0020 | |
| Antibody | Anti-SOX11 (Rabbit monoclonal) | Abcam | Clone EPR8191(2) Cat# ab78078 | WB: (1:1000) |
| Antibody | Anti-SOX11 (Mouse monoclonal) | EBioscience | SOX11-C1 Cat# 50-9773-82 | IF: (1:200) |
| Antibody | Anti-TUBB3 (Mouse monoclonal) | Abcam | Clone 2G10 Cat# ab78078 | WB: (1:1000) IF: (1:100) |
| Antibody | Anti-MEX3A (Rabbit polyclonal) | Abcam | Cat# ab79046 | WB: (1:1000) IF: (1:100) |
| Antibody | Anti-CD24 (Mouse monoclonal) | Creative Biolabs | Cat# SWA11 | WB: (1:1000) |
| Antibody | Anti-VIMENTIN (Rabbit monoclonal) | Abcam | Cat# Ab92547 (EPR3776) | WB: (1:1000) IF: (1:400) |
| Antibody | Anti-RCOR2 (CoREST2) (Rabbit polyclonal) | Abcam | Cat# ab37113 | WB: (1:1000) |
| Antibody | Anti-N-CADHERIN (Rabbit monoclonal) | Cell Signaling | Cat# 13116 | WB: (1:1000) IF: (1:50) |
| Antibody | Anti-E-CADHERIN (Mouse monoclonal) | BD Bioscience | Clone 36 Cat# 610182 | WB: (1:1000) IF: (1:200) |
| Antibody | Anti-EPCAM (Rabbit monoclonal) | Cell Signaling | Clone D1B3 Cat# 2626 | WB: (1:1000) |
| Antibody | Anti-RRM2 (R2) (Mouse monoclonal) | Santa Cruz | Clone A-5 Cat# sc-398294 | WB: (1:1000) |
| Antibody | Anti-SURVIVIN (Rabbit monoclonal) | Cell Signaling | Clone 71G4B7 Cat# 2808 | WB: (1:1000) |
| Antibody | Anti-GAPDH (Rabbit monoclonal) | Cell Signaling | Clone D16H11 Cat# 5174 | WB: (1:5000) |
| Antibody | Anti-LAMINB1 (Rabbit polyclonal) | Abcam | Cat# ab16048 | WB: (1:1000) |

*Continued on next page*

*Continued*

| Reagent type (species) or resource | Designation | Source or reference | Identifiers | Additional information |
|---|---|---|---|---|
| Antibody | Anti-β-TUBULIN (Mouse monoclonal) | Sigma | Cat# T4026 | WB: (1:1000) |
| Antibody | Anti-β-ACTIN (Mouse monoclonal) | Cell Signaling | Clone 8H10D10 Cat# 3700 | WB: (1:1000) |
| Antibody | Anti-β-K5 (Rabbit polyclonal) | Biolegend | Cat# PRB-160P | IF: (1:200) |
| Antibody | Anti-β-K14 (Rabbit polyclonal) | Biolegend | Cat# PRB-155P | IF: (1:200) |
| Antibody | Anti-β-CD24 (Mouse monoclonal) | Invitrogen/ Thermofisher | Cat# SN3 | IF: (1:50) |
| Antibody | Anti-β-SMA (Rabbit monoclonal) | Invitrogen/ Thermofisher | EPR5368 Cat# Ab202509 | IF: (1:50) |
| Antibody | Anti-CD24–PE–Cy7 (Mouse monoclonal) | BD Bioscience | Cat# 561646 | Flow cytometry: (1:50) |
| Antibody | Anti-CD44–APC (Mouse monoclonal) | BD Bioscience | Cat# 559942 | Flow cytometry: (1:50) |
| Recombinant DNA reagent | pInducer21 (Plasmid) | Addgene | Cat# 46948 | |
| Recombinant DNA reagent | pInducer13 (Plasmid) | Addgene | Cat# 46936 | |
| Recombinant DNA reagent | Firefly Luciferase 2 lentiviral particles | Amsbio | Cat# LVP325 | |
| recombinant DNA reagent | pLV-mCherry (Plasmid) | Addgene | Cat# 36084 | |
| Sequence-based reagent | TaqMan probe SOX11 | Thermofisher Scientific | Hs00846583_s1 | |
| Sequence-based reagent | TaqMan probe TUBB3 | Thermofisher Scientific | Hs00801390_s1 | |
| Sequence-based reagent | TaqMan probe MEX3A | Thermofisher Scientific | Hs00863536_m1 | |
| Sequence-based reagent | TaqMan probe GPC2 | Thermofisher Scientific | Hs00415099_m1 | |
| Sequence-based reagent | TaqMan probe MAPK4 | Thermofisher Scientific | Hs00969401_m1 | |
| Sequence-based reagent | TaqMan probe LBH | Thermofisher Scientific | Hs00368853_m1 | |
| Sequence-based reagent | TaqMan probe SERPINA3 | Thermofisher Scientific | Hs00153674_m1 | |
| Sequence-based reagent | TaqMan probe OLFM2 | Thermofisher Scientific | Hs01017934_m1 | |
| Sequence-based reagent | TaqMan probe N-CADHERIN | Thermofisher Scientific | Hs00983056_m1 | |
| Sequence-based reagent | TaqMan probe ST6GALNAC5 | Thermofisher Scientific | Hs05018504_s1 | |
| Sequence-based reagent | TaqMan probe GAPDH | Thermofisher Scientific | Hs02786624_g1 | |
| Peptide, recombinant protein | Animal-Free Recombinant Human EGF | Peprotech | Cat# AF-100-15 | |
| Commercial assay or kit | RNAeasyPlus Micro kit | Qiagen | Cat# 74034 | |
| Commercial assay or kit | RNAClean and concentrator-5 | Zymo Research | Cat# R1013 | |

*Continued on next page*

Continued

| Reagent type (species) or resource | Designation | Source or reference | Identifiers | Additional information |
|---|---|---|---|---|
| Commercial assay or kit | Agilent RNA Pico kit | Agilent Technologies | Cat# 5067-1513 | |
| Commercial assay or kit | QuantiTect Reverse Transcription kit | Qiagen | Cat# 205311 | |
| Commercial assay or kit | TaqMan Gene Expression Master Mix | Thermofisher Scientific | Cat# 4369016 | |
| Commercial assay or kit | Aldefluor assay | StemCell Technologies | Cat# 01700 | |
| Commercial assay or kit | PKH26 Dye Solution | SIGMA | Cat# MINI26 | |
| Commercial assay or kit | Tumour dissociation kit, human | Miltenyi | Cat# 130-095-929 | |
| Chemical compound, drug | Doxycycline hyclate | Sigma | Cat# D9891 | |
| Chemical compound, drug | NeuroCult SM1 without vitamin A | StemCell Technologies | Cat # 05731 | |
| Chemical compound, drug | Methylcellulose | R&D Systems | Cat # HSC002 | |
| Chemical compound, drug | Cell titer-Glo | Promega | Cat # G7572 | |
| Chemical compound, drug | Collagen I, High Concentration, Rat Tail | Corning | Cat # 354249 | |
| Chemical compound, drug | XenoLight D-Luciferin Potassium Salt | Perkin Elmer | Cat # 122799 | |
| Chemical compound, drug | Lipofectamine 2000 | Invitrogen | 11668019 | |
| Chemical compound, drug | Lipofectamine RNAiMAX | Invitrogen | 13778075 | |
| Software, algorithm | PRISM | Graphpad | | |
| Software, algorithm | BD FACS Diva software | BD Bioscience | | |
| Software, algorithm | Image J | National Institutes of Health (NIH) | | |
| Other | EVOS FL microscope | Thermofisher Scientific | | |
| Other | Confocal microscope | Leica | Model TCS-SP2 | |
| Other | Celigo cytometer | Nexcelom | | |
| Other | 96-well ultra-low-attachment plates | Corning | Cat # 7007 | |
| Other | Luminescence plate reader | Perkin Elmer | Victor X5 58 | |
| Other | FACS | BD Bioscience | FACSAriaIII | |
| Other | Flow cytometer | BD Bioscience | BD FACS LSRII | |
| Other | Stereotaxic frame | Stoelting | | |
| Other | IVIS Lumina imaging systems | Perkin Elmer | | |
| Other | gentleMACS Octo Dissociator with Heaters | Perkin Elmer | | |
| Other | DAPI | Sigma | IF: 1:5000 FC/FACS: 1:5000 | |

## Cell culture

DCIS.com-Luc cells were generated by transducing cells with lentiviral expression particles for firefly luciferase 2 (LVP325; Amsbio, Abingdon, UK). DCIS.com-Luc-mCherry were created by lentiviral transduction of the mCherry sequence in the DCIS.com-Luc cells. Supplementary material (*Supplementary file 6*) provides details and sources of cell lines and media used. All cell lines were tested and were mycoplasma-free. The DCIS.com have been extensively profiled in *Maguire et al., 2016* and we confirmed PIK3CA mutation status by PCR and Western blotting. Cell lines were authenticated by STR profiling (Eurofins).

## Expression vectors

The SOX11 coding sequence (GENEID: 6664) from clone HsCD00295480 19 in the pENTR223.1 plasmid (DNASU) 20 was subcloned into pInducer21 (ORF-EG) plasmid gift from Stephen Elledge and Thomas Westbrook (Addgene plasmid # 46948; http://n2t.net/addgene: 46948; RRID:Addgene_46948) (*Meerbrey et al., 2011*). pInducer13-shRNA and pInducer13-NS shRNA were made by subcloning the specific shRNA174 for SOX11 and the NS shRNA from pGIPZ plasmids from Horizon into pInducer13.

## RNA isolation

RNA from cells grown in 2D for 48 hr in presence or absence of 1 µM doxycycline (DOX) and from spheroids treated for 2 or 5 days (DOX 2 days or DOX 5 days) or not treated with DOX (DOX 0 Days) (n = 3 for each time point) was isolated with an RNAeasyPlus Micro kit (74034; Qiagen, Manchester, UK) and DNase treatment. RNAClean and concentrator-5 (Zymo Research, Irvine, CA) were used. RNA concentration and purity were determined with a Qubit fluorometer (Invitrogen, Carlsbad, C) and a nanodrop spectrophotometer. RNA integrity number was measured with a bioanalyzer and an Agilent RNA Pico kit (Agilent Technologies, Cheshire, UK).

## cDNA synthesis and qPCR

One microgram of each RNA sample was reverse transcribed with QuantiTect Reverse Transcription kit (Qiagen, Manchester, UK) in a final volume of 20 µl. cDNA was diluted ten times for subsequent quantitative polymerase chain reaction (qPCR) analysis, as described previously (*Oliemuller et al., 2017*), with the probes and methods listed in supplementary material (*Supplementary file 7*).

## Western blotting

Western blotting was performed as previously described (*Zvelebil et al., 2013*). Details of the antibodies used are provided in supplementary material (*Supplementary file 8*).

## Immunohistochemistry (IHC)

IHC was performed on formalin-fixed paraffin embedded samples. Samples were stained with antibodies as described previously (*Oliemuller et al., 2017*).

## Immunofluorescence

Antibodies and staining protocols are detailed in supplementary material (*Supplementary file 9*). EVOS fluorescence microscope was used for imaging. Confocal images were captured with a Leica Microsystems (Cambridge, UK) TCS-SP2 confocal microscope.

## Spheroid formation

Five thousand cells, untreated or treated with 1 µM doxycycline (DOX) for 48 hr, were plated per well in 96-well ultra-low-attachment plates (Corning 7007, Corning, NY, USA) in media containing DOX or not. After 24 hr, when the spheroids were formed, new media containing 1 µM doxycycline was added for 48 hr to spheres formed in absence of DOX for 2 days (DOX 2 days) or spheres that were formed in presence of DOX to a total of 5 days (DOX 5 days). Media without doxycycline was added to the control spheroids (No DOX). Images were obtained with a Celigo cytometer (Nexcelom, Manchester, UK).

## Colony-formation assays

DCIS.com cells were plated at 250 per well in six-well (Falcon F3046, Corning, NY) plates. After 7 days, plates were stained with 0.2% crystal violet dissolved in 20% methanol in PBS. Area was measured, and the percentage relative to number of cells plated was calculated.

For single-cell colony assays and single-cell mammosphere assays, cells were FACS sorted and a single living cell plated per well in a 96-well or ultra-low attachment plate respectively. After 14 days, plates were analysed in a Celigo cytometer (Nexcelom, Manchester, UK).

For mammosphere assays, 5000 DCIS cells/ml were plated in low-attachment six-well plates (Corning 3471) and incubated in medium supplemented with 2% NeuroCult SM1 without vitamin A (StemCell Technologies) and 0.65% methylcellulose (R and D Systems, Abingdon, UK). After 14 days, wells were scanned with a Celigo cytometer. Mammosphere-forming efficiency was calculated by dividing the number of mammospheres by the number of cells plated per well.

## Cell viability assays

Three thousand cells per well were plated in 96-well plates (655098; Greiner Bio-one, Stonehouse, UK) or in ultra-low attachment plates for 24 hr before starting the experiments. CellTiter-Glo (Promega, Southampton, UK) was used according to the manufacturer's protocol. Luminescence was measured with a Victor X5 58 plate reader (Perkin-Elmer, Seer Green, UK). In DCIS.com cells, CellTiter-Glo assays were performed at the time of adding the doxycycline and after 1, 2 and 3 days. When cells were transfected with siRNAs, the transfection was done overnight and next day (day 1), full supplemented media was added. CellTiter-Glo assays were performed on days 1 and 5 to allow time for the siRNAs to knock down the genes of interest. CAL-148 cells stably transduced with shRNAs were measured at 0, 3 and 7 days, since this cell line has a long doubling time.

## Invasion assays

Three days after plating 5000 DCIS.com cells, spheroids were embedded in collagen I (354249; Corning) at 2.2 mg/ml diluted in medium. Complete medium was added on top after 1 hr. Images were acquired at this time and after 48 hr with a Celigo cytometer. The total area of matrix invaded by cells was calculated with ImageJ after marking of the area manually. For BT-20 and HCC1187 cells, the spheres used in the invasion assays were formed at the same time that they were transfected with the *TUBB3* or *MEX3A* specific siRNAs in Opti-MEM media supplemented with 10% FBS. After 24 hr, this media was changed for the normal media of this cells. The cells were invaded in collagen 72 hr after plating.

## Tranfections and siRNA

BT-20, HCC1187 and CAL-148 cells were transfected with 50 pmol of each *TUBB3* or *MEX3A* siRNA (siGENOME SMARTpool and four individual siRNAs), control nontargeting siRNA (Thermo Scientific, Waltham, MA) by using Lipofectamine RNAiMAX (Invitrogen, Life Technologies Corporation, Carlsbad, CA) in Opti-MEM media supplemented with 10% FBS (Gibco, Life Technologies Corporation, Carlsbad, CA) media according to the manufacturer's instructions for 16 hr in a six-well or 96-well plate, depending on the experiment, and then incubated with complete media. For sphere transfections, 5000 cells per well were incubated with 2.5 pmol of each specific siRNA in 150 µl of normal media per well in an ultra-low attachment plate for 16 hr. The next day, media was replaced with fresh media.

Flow cytometry analyses and fluorescence-activated cell sorting (FACS) Aldehyde dehydrogenase (ALDH) activity was measured with the Aldefluor assay (StemCell Technologies, Cambridge, UK) as described before (*Oliemuller et al., 2017*). Cells were also co-stained with Aldefluor and anti-CD24–PE–Cy7 (561646) (1:100) and anti-CD44–APC (559942) (1:20) (BD Biosciences, Oxford, UK). A BD FACS LSRII flow cytometer was used and samples were analysed with BD FACS Diva software (BD Biosciences). mCherry+ DCIS cells from the tumours in the tail vein xenograft experiments were sorted by a FACSAriaIII (BD). FACSAriaIII was used for clonogenic and mammosphere assays that required plating single cells per well. Living and dead cells were distinguished with DAPI 1:5000. Cell cycle analysis was described previously (*Zvelebil et al., 2013*).

## PKH staining

two $\times 10^{-6}$ DCIS-LacZ or DCIS-SOX11 cells were resuspended in Diluent C and stained with PKH26 Dye Solution to a final concentration of 5 mM following the manufacturer's protocol (MINI26, Sigma). After confirming by flow cytometry that the 100% of the population stained, spheres with 5000 cells were formed in ultra-low attachment 96-well plates.

## Animal experiments

All animal work was carried out under UK Home Office project and personal licenses following local ethical approval from The Institute of Cancer Research Ethics Committee and in accordance with local and national guidelines. For xenograft tumour assays, DCIS.com labelled with Luc2-mCherry and stably transduced with empty vector (EV) control or SOX11 (in pInducer21 backbone) were resuspended in PBS for implantation into female NSG-Foxn1$^{null}$ mice. $1.0 \times 10^6$ cells/site in both sides were injected in each mammary fat pad number 4 and $2.0 \times 104$ cells/site into the mammary duct via the nipple of mammary gland number four as previously described (*Oliemuller et al., 2017*).

Intracranial injections were performed using a stereotaxic frame by injecting $1 \times 10^{-5}$ cells (Stoelting, Wood Lane, IL, USA) into the striatum (2 mm right from the midline, 2 mm anterior from bregma, 3 mm deep). Six NSG-Foxn1null mice were used per for each condition.

For tail vein injections, the mice were placed into a hot box set at 38°C for up to 5 min, to dilate the tail veins. The mice were then placed into a restrainer, the lateral tail vein identified, and $2.5 \times 10^5$ cells were slowly infused (over 30 s) through the tail vein using a venoflux 25 g butterfly needle. Before treatment, mice used were randomised in groups based on their weight.

For the doxycycline induction experiments using iSOX11 or iEV, a week before the injections, the animals injected with transduced cells were separated into two cohorts and maintained with or without chow containing doxycycline (0.2 g/kg, 0.625, 1.250 or 2.0 g/kg from Envigo) for the duration of the experiment.

For in vivo imaging, mice were injected with 200 µl of luciferin (XenoLight D-Luciferin Potassium Salt, Perkin Elmer, 30 mg/mL). After 5 min, the mice were anaesthetised with isoflurane and the bioluminescence was measured at least at three different time points on an IVIS Lumina imaging systems (Perkin-Elmer). For ex vivo imaging, the organs were resected and imaged for 3 min to detect any signal. In the analysis, identical square regions of interest (ROI) were drawn around tumours to measure total and average bioluminescence signal.

Tumour dissociation iSOX11 or iEV cells from brain, bone and lung metastasis were isolated with a cell dissociation kit following the manufacturer instructions (Miltenyi Biotec, Bergisch Gladbach, Germany) and using the gentleMACS Octo Dissociator with Heaters and its 37°C_h_TDK_3 protocol. To select human cells and discard murine cells, the result of this dissociation was FACS sorted in a FACSAriaIII and mCherry+ cells were selected and grown in normal DCIS media.

RNA sequencing cDNA library preparation was carried out at Oxford Genomics Centre, The Wellcome Trust Centre for Human Genetics using PolyA+ RNA enrichment method for total RNA from cultured cells. mRNA fraction was selected from the total RNA before conversion to cDNA. Second strand cDNA synthesis incorporated dUTP. The cDNA was end-repaired, A-tailed and adapter-ligated. Prior to amplification, samples underwent uridine digestion. The prepared libraries were size selected, multiplexed and quality checked before paired end sequencing over four lanes of a flow cell.

Sequence files were trimmed by the use of trim_galore (http://www.bioinformatics.babraham.ac.uk/projects/trim_galore/) with default settings. Trimmed data were separately mapped to the GRCh38 and GRCm38 genome assemblies by the use of hisat2 (v2.0.5) with options --sp 1000,1000 -- omixed--no-discordant, and were filtered to remove non-primary alignments. Species-specific read sets were generated by removing any read that produced a valid alignment in both human and mouse from the results for both species. The remaining data were imported into SeqMonk (http://www.bioinformatics.babraham.ac.uk/projects/seqmonk/) with a filter of mapping quality (MAPQ) score $\geq 20$. Reads were quantified over the transcript set from Ensembl v78 with annotated mis--spliced, pseudogene and unannotated transcripts removed. Initial quantification was raw read counts from the opposing strand to the transcript, with all exons for each gene being collated into a single measure. This allowed gene-level differential expression to be assessed by the use of DESeq2

([https://bioconductor.org/packages/release/bioc/html/DESeq2.html](https://bioconductor.org/packages/release/bioc/html/DESeq2.html)), with a cutoff of a false discovery rate of <0.05. Subsequent visualisation was performed by requantifying expression as log2 fragments per million reads of library. RNA sequencing files were submitted to ArrayExpress.

## Survival analysis

The prognostic importance of *SOX11* mRNA expression was assessed by the use of survival data using the cBioPortal for Cancer Genomics ([http://cbioportal.org](http://cbioportal.org)) (*Cerami et al., 2012*; *Gao et al., 2013*). Data obtained from The Cancer Genome Atlas ([https://www.cancer.gov/tcga](https://www.cancer.gov/tcga)) was examined by use of the Kaplan–Meier Plotter survival analysis tool ([http://kmplot.com](http://kmplot.com)) and METABRIC (*METABRIC Group et al., 2012*), and statistical significance was determined with the Wald test.

## Statistical analysis

The data in the graphs are presented as mean and standard deviation, unless specified otherwise. Experiments were analysed with a two-tailed Student's t-test with a confidence interval of 95% when the number of groups equalled 2, or with a parametric ANOVA and *post hoc* test when the number of groups was >2, unless otherwise specified. p-value$\leq$0.0001 is considered as extremely significant (****), p$\leq$0.001 as highly significant (***), p$\leq$0.01 as very significant (**), p$\leq$0.05 as significant (*), and p>0.05 as not significant (ns), respectively.

# Acknowledgements

We thank Breast Cancer Now for funding this work as part of Programme Funding to the Breast Cancer Now Toby Robins Research Centre. This work was supported by a grant from CRUK (CRUK: A21855). This work was supported by grants from the Spanish Ministry of Education and Science (SAF2017-84934-R, to MV) and the Government of the Autonomous Community of the Basque Country, the Department of Industry, Tourism and Trade (Elkartek: KK-2018/00054, IA-R and MV). We acknowledge the role of the Breast Cancer Now Tissue Bank in collecting and making available the normal breast samples used in the generation of this publication. We thank the Oxford Genomics Centre at the Wellcome Centre for Human Genetics (funded by Wellcome Trust grant reference 203141/Z/16/Z) for the generation and initial processing of the sequencing data.

# Additional information

## Funding

| Funder | Grant reference number | Author |
|---|---|---|
| Breast Cancer Now | Programme Funding to the Breast Cancer Now Toby Robins Research Centre | Erik Oliemuller<br>Richard Newman<br>Siu Man Tsang<br>Gareth Muirhead<br>Farzana Noor<br>Syed Haider<br>Beatrice A Howard |
| Cancer Research UK | CRUK: A21855 | Shane Foo |
| Spanish Ministry of Education and Science | SAF2017-84934-R | Maria dM Vivanco |
| Government of the Autonomous Community of the Basque Country, the Department of Industry, Tourism and Trade | KK-2018/00054 | Iskander Aurrekoetxea-Rodríguez<br>Maria dM Vivanco |

The funders had no role in study design, data collection and interpretation, or the decision to submit the work for publication.

## Author contributions

Erik Oliemuller, Data curation, Formal analysis, Investigation, Writing - review and editing; Richard Newman, Siu Man Tsang, Investigation, Writing - review and editing; Shane Foo, Maria dM Vivanco,

Investigation, Methodology; Gareth Muirhead, Syed Haider, Formal analysis, Methodology; Farzana Noor, Iskander Aurrekoetxea-Rodríguez, Investigation; Beatrice A Howard, Conceptualization, Supervision, Funding acquisition, Investigation, Writing - original draft, Project administration, Writing - review and editing

### Author ORCIDs
Erik Oliemuller  http://orcid.org/0000-0002-4506-0504
Beatrice A Howard  https://orcid.org/0000-0002-9162-0314

### Ethics
Animal experimentation: All animal work was carried out under UK Home Office PPL number: PB0FA698C (BAH Project licence holder) and personal licenses following local ethical approval of all protocols from The Institute of Cancer Research Ethics Committee and in accordance with local and national guidelines. All surgery was performed under isoflurane anesthesia with appropriate analgesia, and every effort was made to minimise suffering.

### Decision letter and Author response
Decision letter https://doi.org/10.7554/eLife.58374.sa1
Author response https://doi.org/10.7554/eLife.58374.sa2

## Additional files

### Supplementary files
• Supplementary file 1. RNA-seq results when SOX11 is induced in cells grown in 2D, 3D for 2 days or 3D for 5 days.

• Supplementary file 2. Gene ontology analysis of the genes differentially expressed in the 3 RNA-seq datasets when SOX11 is induced in cells grown in 2D, 3D for 2 days or 3D for 5 days.

• Supplementary file 3. Genes differentially expressed the 3 RNA-seq datasets when SOX11 is induced in cells grown in 2D, 3D for 2 days or 3D for 5 days.

• Supplementary file 4. Expression values of the genes from stem cell quiescence signature in the three datasets obtained in DCIS when SOX11 is induced in cells grown in 2D, 3D for 2 days or 3D for 5 days.

• Supplementary file 5. Co-occurrence and correlation of *MEX3A* RNA levels with cell cycle related genes in TCGA breast cancer dataset.

• Supplementary file 6. Cell lines and culture media.

• Supplementary file 7. qPCR probes.

• Supplementary file 8. Antibodies used for western blotting.

• Supplementary file 9. Antibodies used for IF and IHC.

• Transparent reporting form

### Data availability
Sequencing data have been deposited in ArrayExpress as accession E-MTAB-9108. All data generated or analysed during this study are included in the manuscript and supporting files.

The following dataset was generated:

| Author(s) | Year | Dataset title | Dataset URL | Database and Identifier |
|-----------|------|---------------|-------------|-------------------------|
| Oliemuller E, Howard BA | 2020 | RNA-seq of DCIS-pInducer21-SOX11 cells grown in 2D and 3D | https://www.ebi.ac.uk/arrayexpress/experiments/E-MTAB-9108/ | ArrayExpress, EBI |

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
