## [Decision Letter]

**Acceptance summary:**

This work clarifies the role of *SOX11* in breast cancer metastasis to the brain and bone via regulation of *TUBB3* and *MEX3A* and effects on tumor cell quiescence and viability. The work has significance for understanding why *SOX11* is associated with metastasis and poor clinical outcomes in breast cancer patients.

**Decision letter after peer review:**

Thank you for submitting your article "*SOX11* promotes epithelial/mesenchymal hybrid state and alters tropism of invasive breast cancer cells" for consideration by *eLife*. Your article has been reviewed by three peer reviewers, one of whom is a member of our Board of Reviewing Editors, and the evaluation has been overseen by Richard White as the Senior Editor. The reviewers have opted to remain anonymous.

The reviewers have discussed the reviews with one another and the Reviewing Editor has drafted this decision to help you prepare a revised submission.

The current work builds on previous work by these authors identifying a role for *SOX11* in promoting invasive and metastatic properties to breast epithelial cells. Here they generate a new DOX-inducible cell line in DCIS.com cells to turn *SOX11* on. Using this system, they show that *SOX11* induces expression of mesenchymal markers, slows growth of cells and spheroids, increases invasiveness and alters metastatic tropism and identifies *MEX3A* as a *SOX11* mediator in invasiveness and metastasis.

Summary:

Overall, the authors found the work to be interesting and novel but requiring various revisions to make the conclusions convincing, as noted below. Also, reviewers suggest that the language be tempered on what is being claimed since the data is not fully supportive of those claims.

Essential revisions:

1) Controls are lacking or inappropriate in some figures and need to be amended – for example, please justify the use of the pooled control lines in Figure 3, provide blots for *SOX11* in all control lines, provide a blot for *TUBB3* knockdown etc.

2) Please justify use of the specific time points used for "proliferation" assays.

3) Please address reviewer #2's concern about IVIS data.

4) Most importantly, please moderate claims about quiescence, multipotency and stemness in line with data presented and with particularly attention to the reviewers' comments.

Reviewer #1:

The current work builds on previous work by these authors identifying a role for *SOX11* in promoting invasive and metastatic properties to breast epithelial cells. Here they generate a new DOX-inducible cell line in DCIS.com cells to turn *SOX11* on. Using this system, they show that *SOX11* induces expression of mesenchymal markers, slows growth of cells and spheroids, increases invasiveness and alters metastatic tropism and identifies *MEX3A* as a *SOX11* mediator in invasiveness and metastasis.

Overall, the data presented, while related to previous work, are an advance on that previously reported work. There are a few concerns that should be addressed:

1) The authors claim that the *iSOX11* lines shows elevated *SOX11* expression compared to lines worked with previously and as a result *SOX11* is found at sub-cellular compartments where it would not normally be found. Given concern that such supra-physiological levels may cause abnormal phenotypes, could the authors not have examined *SOX11* expression in additional clones to find lines with more physiological levels of *SOX11*?

2) The authors refer to "proliferation" in Figure 2, when in fact they are essentially counting cell number which is also affected by rates of cell death (which they do not measure). They should either adjust the use of the term "proliferation" and use instead "cell growth" or they should measure and exclude increased cell death as the cause of fewer cells and smaller spheroids.

3) The authors need to show a western blot for *TUBB3* knockdown in Figure 7.

4) The extent of *MEX3A* knockdown in Figure 8 is variable and approximately half in best case siRNA experiments but effects of the knockdown, including invasion etc do not seem to correlate with the level of knockdown.

Reviewer #2:

This manuscript by Oliemuller et al. details the role of *Sox11* in breast cancer and metastasis. It extends on earlier work published by this group (Oliemuller et al., 2017), using a new inducible vector to overexpress *Sox11*, and provides new data relating to metastasis. One of the most interesting findings is the predilection of the *iSOX11* DCIS.com cell line to metastasize to brain. The CSC section should be downplayed, as this was not examined further in the study. Also, CD24 has been reported by many groups to be low in breast CSCs, while CD44 is elevated.

The work on EMT genes seems to form a separate stream of work. For the EMT section in the Results, a candidate approach was taken, but a more systematic RNA sequencing approach would provide further data. Does gene expression vary with the assay performed (collagen gel, 2D or spheroid), and if so, this would confound interpretation. The quiescence angle is not clear and without further data, senescence or another process cannot be excluded.

Reviewer #3:

This is a clear well-written and beautifully presented paper on an important topic of high relevance for our understanding of aggressive forms of human breast cancer. The authors thoroughly dissect a *Sox11*, *TUBB3* and *Mex3A* pathway and demonstrate control by these genes of breast cancer cell phenotype, proliferation and most importantly metastatic behavior and tropism. They trace this regulatory pathway in a scholarly fashion by upregulating and downregulating expression of these genes in DCIS.com a relevant breast cancer cell line to show that Sox 11 enhances breast cancer cell quiescence and viability, augments expression of both basal and luminal keratins as well as mesenchymal factors vimentin and N-cadherin and reduces and alters localization of epithelial adhesion molecules, E-cadherin and EpCam. This manuscript clarifies the mechanism behind *SOX11* association with brain and bone metastatic tropism and consequent poor outcome in humans. I believe the paper is acceptable as is and will be of interest to the readership of *eLife*.

---

## [Author Response]

Essential revisions:1) Controls are lacking or inappropriate in some figures and need to be amended – for example, please justify the use of the pooled control lines in Figure 3, provide blots for SOX11 in all control lines, provide a blot for TUBB3 knockdown etc.

We have clarified the controls used comparison in the legend for Figure 3. To control for effects of DOX treatment on DCIS.com cells, we have subtracted the values of the EV cells treated with DOX compared to non-treated EV cells from the induction observed in the *iSOX11* cells treated with DOX from uninduced *iSOX11* cells.

We have provided the appropriate controls which we had inadvertently omitted from the original submission. Blots of *SOX11* are shown in Figure 3G and we have provided a blot for *TUBB3* knockdown Figure 7A and blots for *MEX3A* knockdown are included in Figure 8—figure supplement 2.

2) Please justify use of the specific time points used for "proliferation" assays.

We have changed the reference to proliferation assays to cell growth assays in Figures 2A, B, 6C, 7A, 8C.

Since siRNA-mediated knockdown is not efficient until 48 hours after transfection, we do the assays at 5 days. CAL-148 cells grow very slowly so cell growth assays were performed on day 3 and 7. We have explained the use of timepoints used in the cell growth assays in the figure legends for Figures 6C, 7B, 8C.

3) Please address reviewer #2's concern about IVIS data.

We appreciate the concern about the IVIS data and have shown another image that show the cells on all of the edges of the control tumours display bioluminescence, but less so in the central regions. We now routinely image mice with mammary tumours lying on their side so that the whole tumour is effectively imaged. The larger tumours were filled with fluid. H&Es are now shown in Figure 2H to show that control tumours display central necrosis.

4) Most importantly, please moderate claims about quiescence, multipotency and stemness in line with data presented and with particularly attention to the reviewers' comments.

We have revised and toned down our claims about quiescence, multipotency and stemness.

Reviewer #1:The current work builds on previous work by these authors identifying a role for SOX11 in promoting invasive and metastatic properties to breast epithelial cells. Here they generate a new DOX-inducible cell line in DCIS.com cells to turn SOX11 on. Using this system, they show that SOX11 induces expression of mesenchymal markers, slows growth of cells and spheroids, increases invasiveness and alters metastatic tropism and identifies MEX3A as a SOX11 mediator in invasiveness and metastasis.Overall, the data presented, while related to previous work, are an advance on that previously reported work. There are a few concerns that should be addressed:1) The authors claim that the iSOX11 lines shows elevated SOX11 expression compared to lines worked with previously and as a result SOX11 is found at sub-cellular compartments where it would not normally be found. Given concern that such supra-physiological levels may cause abnormal phenotypes, could the authors not have examined SOX11 expression in additional clones to find lines with more physiological levels of SOX11?

We would like to emphasize that *SOX11* is not detected in normal breast epithelial cells. We decided to make the inducible model since the previous constitutive CMV-driven *SOX11* was expressed at very low levels and was not detected at the levels we observed in DCIS case samples (Oliemuller, 2017). Staining with *SOX11* by IHC shows comparable expression levels in the *iSOX11* mouse tumours (Figure 2—figure supplement 4) and in both pure DCIS and mixed cases samples (Figure 1C) where predominantly nuclear *SOX11* is detected. In neither case, is cytoplasmic *SOX11* staining obvious. The staining of *iSOX11* cells by IF with *SOX11* (Figure 1B) shows predominantly nuclear staining and very little is detected in the cytoplasm. In addition, our analysis of *SOX11* expression levels in breast cancers shows that some clinical samples express very high levels of *SOX11* whilst others express lower levels of *SOX11,* in line with our two models. We have not analysed the subcellular compartment in which *SOX11* is expressed by Western blotting in breast cancer samples.

2) The authors refer to "proliferation" in Figure 2, when in fact they are essentially counting cell number which is also affected by rates of cell death (which they do not measure). They should either adjust the use of the term "proliferation" and use instead "cell growth" or they should measure and exclude increased cell death as the cause of fewer cells and smaller spheroids.

We have changed the use of the term “proliferation” to “cell growth”

3) The authors need to show a western blot for TUBB3 knockdown in Figure 7.

We have included the western blot showing *TUBB3* knockdown in Figure 7A.

4) The extent of MEX3A knockdown in Figure 8 is variable and approximately half in best case siRNA experiments but effects of the knockdown, including invasion etc do not seem to correlate with the level of knockdown.

Levels of *MEX3A* knockdown in Figure 8 is similar with the various siRNAs, and does not correlate, particularly with respect to effects on invasion. We have noted this in the subsection “*MEX3A*, a novel potential *SOX11* downstream effector, regulates cell growth and E/M state of ER- breast cancer cells”.

Reviewer #2:This manuscript by Oliemuller et al. details the role of Sox11 in breast cancer and metastasis. It extends on earlier work published by this group (Oliemuller et al., 2017), using a new inducible vector to overexpress Sox11, and provides new data relating to metastasis. One of the most interesting findings is the predilection of the iSOX11 DCIS.com cell line to metastasize to brain. The CSC section should be downplayed, as this was not examined further in the study. Also, CD24 has been reported by many groups to be low in breast CSCs, while CD44 is elevated.

Cancer stem cell phenotypes are not fixed (see Liu reference that we cite in the Discussion). CD24 has indeed been reported by many groups to be low in breast CSCs in past studies. It has recently been suggested that breast cancers that have been classified as CD24 low by FACs may have nuclear CD24 localisation (see Deux citation in the Results). It is clear that CD24+ breast cancer cells have a role in breast cancer and present treatment opportunities based on immunotherapy (see Barkal citation). The role of CD24 as a cancer stem cell marker is not entirely clear in our study so we have downplayed the CSC section, as suggested.

The work on EMT genes seems to form a separate stream of work. For the EMT section in the Results, a candidate approach was taken, but a more systematic RNA sequencing approach would provide further data. Does gene expression vary with the assay performed (collagen gel, 2D or spheroid), and if so, this would confound interpretation. The quiescence angle is not clear and without further data, senescence or another process cannot be excluded.

We assessed the expression of established EMT markers (protein expression) by Western blotting and IF, as most researchers do to assess cell state. RNA sequencing was performed (see Supplementary file 1 and Figure 3). In the Results, we have added that RNA-sequencing detected significant upregulation of *CDH2* (*N-CADHERIN*) in both 2D and 3D conditions. Gene expression analysis by RNA sequencing of cells grown in 2D and 3D as spheroids did not detect significant changes in *CDH1* (E-Cadherin) or *VIM (Vimentin)* levels. Until robust markers of quiescence are established, and other experimental data are performed, it will not be clear if quiescence, senescence or other process such as dormancy are involved. We have stated this in the Discussion: *MEX3A* may mark a rare mammary stem cell in the human breast that could escape traditional chemotherapy treatments, but this remains to be demonstrated.